# Bridging Tokens and Geometry: Token-wise 3D Supervision for CAD Generation

**Yijia Guan** [1]  **Jianhua Sun** [1]

## Abstract

Computer-Aided Design (CAD) generation is typically formulated as a sequence modeling task over parametric tokens. Recent studies introduce visual information through additional visual inputs or rendering of the final generated programs. However, these methods provide no intermediate visual feedback, hindering the association of individual tokens with their geometric effects. In this work, we propose an Argument-induced 3D Point Loss (A3PL) that maps argument tokens to corresponding 3D points, enabling dense token-wise geometric supervision. To reduce learning complexity and invalid sequences, we further introduce a Grammar-constrained Operator (GCO) that leverages the structured nature of CAD programs to regulate sequence generation. We evaluate our approach on five CAD generation tasks with diverse input modalities, including text, Scalable Vector Graphics (SVG) sketches, point clouds, and CAD sequences. Our approach improves generation accuracy and program validity across different input modalities. Code is available at https://github.com/JumpJumpTigger-GYJ/A3PL.

## 1. Introduction

Computer-Aided Design (CAD) is a fundamental tool for industrial design and manufacturing. Unlike boundary representations (B-rep) widely used in CAD software (Dassault Systèmes, 2024; FreeCAD Development Team, 2024), CAD programs define 3D geometry as a sequence of parametric operations (Xu et al., 2022), with explicit design history and editability (Wu et al., 2021). CAD program sequence modeling offers a promising approach for 3D design. Its sequential and structured nature aligns well with Transformer-based ar-

chitectures (Vaswani et al., 2017). However, unlike standard sequence modeling tasks, the quality of a CAD program is not determined solely by the correctness of the token sequence; it also depends on whether the rendered 3D geometry matches the intended design. Therefore, incorporating visual information during modeling is crucial for achieving geometrically correct results.

Some introduce additional visual modalities, such as images (Doris et al., 2026; Li et al., 2025) or point clouds (Dupont et al., 2024; Xu et al., 2024), to provide complementary geometric information (Figure 1(b)). Other works (Wang et al., 2025a; Guan et al., 2026) render 3D objects from the final generated CAD program and derive geometry-based rewards for training (Figure 1(c)). Despite providing geometric information, these methods still lack token-wise geometric feedback for individual argument predictions.

CAD modeling also suffers from invalid program generation. CAD sequences are highly structured: structural tokens are deterministically derived from preceding tokens and enforce grammar constraints, whereas semantic tokens encode design decisions, such as primitive types and their arguments, which directly determine the generated geometry. Previous works typically do not differentiate structural and semantic tokens during training, which expands the prediction space and increases the likelihood of invalid CAD programs.

As shown in Figure 1(a), we propose an Argument-induced 3D Point Loss (A3PL) to provide intermediate geometric feedback for each argument token in a CAD sequence. Unlike prior works (Figure 1(b)(c)), A3PL lifts each 2D sketch point to its corresponding 3D point, which is fully differentiable and can be conveniently integrated into Transformer-based models. To reduce learning complexity and the invalidity ratio, we introduce a Grammar-constrained Operator (GCO), as shown in Figure 1(a). Under grammar constraints, GCO identifies structural tokens, masks them during loss computation, and deterministically recovers them at inference. This mechanism allows the model to focus on semantic tokens critical to geometry. Our contributions can be summarized as:

- We introduce token-wise 3D supervision for CAD sequence generation, providing direct point-level supervision for each predicted argument and bridging the gap

[1]School of Artificial Intelligence, Shanghai Jiao Tong University, Shanghai, China. Correspondence to: Jianhua Sun <gothic@sjtu.edu.cn>.

*Proceedings of the 43rd International Conference on Machine Learning*, Seoul, South Korea. PMLR 306, 2026. Copyright 2026 by the author(s).

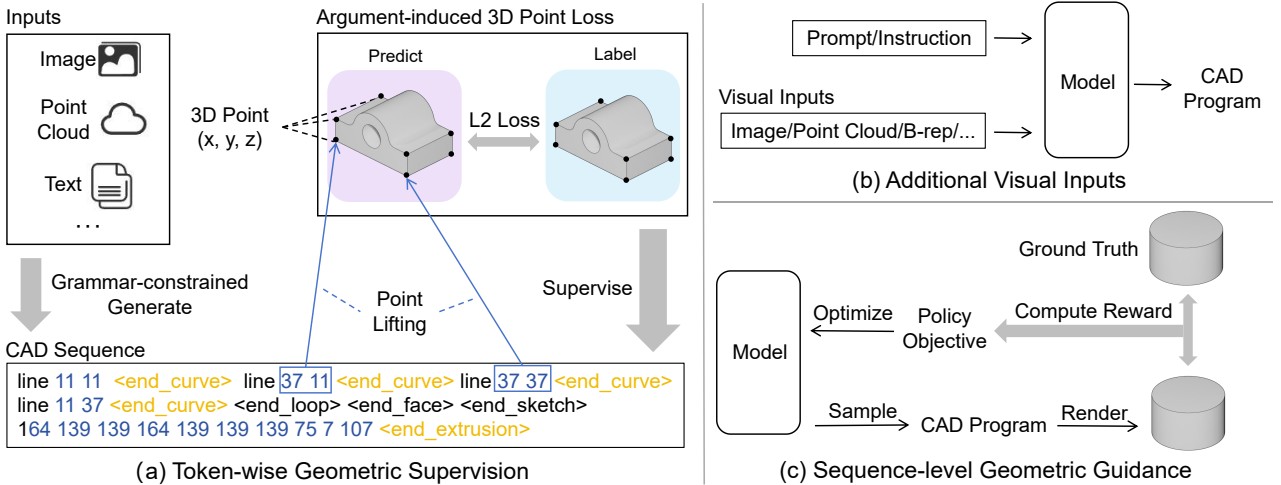

*Figure 1.* Different ways to incorporate visual information into CAD sequence generation. (a) Our method introduces token-wise geometric supervision. The CAD sequence is generated under grammar constraints, and tokens corresponding to 2D sketch points are lifted into 3D points. Argument-induced 3D Point Loss (A3PL) is then employed to provide geometric feedback for each 3D point. (b) A way to incorporate additional visual inputs to provide complementary visual information, but without visual supervision. (c) Another approach applies sequence-level geometric guidance based on reinforcement learning. It renders sampled CAD programs into 3D objects and computes rewards such as Chamfer Distance (CD) and Intersection over Union (IoU) for optimization. However, it lacks token-wise geometric guidance.

between symbolic CAD tokens and their underlying 3D geometry.

- We propose an Argument-induced 3D Point Loss that differentiably lifts sketch and extrusion arguments into 3D space, enabling intermediate geometric feedback during training.

- We propose a Grammar-constrained Operator that masks structural tokens during training and applies grammar-constrained decoding at inference, simplifying learning and improving sequence validity.

- We conduct experiments on five CAD sequence generation tasks with diverse input modalities, including text, Scalable Vector Graphics (SVG) sketches, point clouds, and CAD sequences, demonstrating the generality of our approach across modalities and its effectiveness in generating accurate CAD programs.

## 2. Related Works

**CAD Sequence Modeling.** Early learning-based approaches to CAD generation primarily focused on direct geometric reconstruction, including B-rep modeling (Lambourne et al., 2021), mesh-based generation (Nash et al., 2020), or implicit shape representations (Park et al., 2019). While effective at capturing final geometry, such approaches fail to model the procedural design history essential for interpretable and editable CAD design. Recent works have explored representing CAD models as symbolic programs composed of parametric operations. DeepCAD (Wu et al., 2021) learns CAD models as sequences of sketch-and-extrude operations. CAD-Recode (Rukhovich et al., 2025) translates point clouds into executable CAD Python code. HNC-CAD (Xu et al., 2023) adopts a hierarchical CAD representation with multi-level neural code trees. Our work focuses on sketch–extrude CAD sequences similar to Deep-CAD, as this representation closely mirrors standard CAD design workflows and yields a structured, interpretable CAD program.

Recent studies have further explored fine-grained controllable CAD generation. FlexCAD (Zhang et al., 2025) introduces a unified framework for controllable generation across multiple CAD construction hierarchies. GeoCAD (Zhang et al., 2026) further focuses on local geometry control, allowing users to modify specific local parts according to fine-grained geometric instructions. These works highlight the growing importance of fine-grained control and user-interactive design in CAD generation.

**Visual Information.** Current studies incorporate visual information into CAD sequence generation from two perspectives. Some works introduce additional visual inputs to guide sequence prediction. GACO-CAD (Wang et al., 2025b) augments RGB images with depth and surface normals, and FreeCAD (Lin et al., 2025) fuses multi-view information via a refinement module. However, these methods provide no visual feedback during training. Another line of work (Li et al., 2026; Wang et al., 2025a; Guan et al., 2026) obtains visual feedback through geometry-based re-

wards derived from the rendered 3D objects of the final generated sequences. However, these methods rely on non-differentiable execution and provide sparse, sequence-level feedback, whereas our approach offers dense, token-wise geometric supervision.

**Structured Sequence Generation.** Structured sequence generation often produces invalid outputs due to weak constraints. SD-VAE (Dai et al., 2018) enforces semantic constraints via stochastic lazy attributes during decoding, while grammar-aligned decoding (Park et al., 2024) ensures syntactic validity through adaptive token restriction and approximate lookahead. Type-constrained decoding (Mündler et al., 2025) further enforces well-typedness using type systems and prefix automata, reducing compilation errors. Inspired by these structured generation techniques, we introduce grammar constraints for CAD sequence generation, which also involves highly structured outputs with complex token dependencies.

## 3. Method

### 3.1. CAD Representation & Model Architecture

**CAD Representation.** We adopt a CAD representation largely following (Khan et al., 2024a), with three additional tokens (<line>, <arc>, and <circle>) to denote primitive types. Further details are provided in the supplementary material. For clarity, in all figures, 2D argument tokens are flattened into a one-dimensional form and arranged within the sequence alongside other tokens.

**Model Architecture.** Our method is primarily designed for autoregressive Transformer-based CAD sequence modeling and is consistent with prior works (Khan et al., 2024b;a; Wang et al., 2025a).

### 3.2. Argument-induced 3D Point Loss

In CAD sequence modeling, the lack of intermediate geometric supervision restricts learning to symbolic tokens and motivates our design. We propose Argument-induced 3D Point Loss (A3PL) to inject token-wise geometric feedback during training, as shown in Figure 1(a) and Figure 2. A3PL is motivated by the observation that the geometry of CAD primitives is primarily controlled by their arguments, which implicitly define key geometric points of the resulting solid. By supervising the 3D positions of these predicted points, the model can learn to associate arguments tokens with their underlying 3D geometry.

Figure 2(a) illustrates the two key components involved in point lifting: the sketch sequence, which contains 2D points of primitives, and the extrusion sequence, which specifies the extrusion operation along with its associated scale, rotation, and translation parameters. Assume that a sketch

sequence $S$ and an extrusion sequence $E$ correspond to the same solid, i.e., the shortest subsequence that starts after a <start> or <end_extrusion> token and terminates with a <end_extrusion> token. $S$ defines a set of 2D sketch points $P^{2D} \in \mathbb{R}^{N_p \times 2}$, while $E$ specifies extrusion distances $d^+, d^- \in \mathbb{R}$, scale factor $s \in \mathbb{R}$, rotation matrix $R_{xyz} \in \mathbb{R}^{3 \times 3}$ and translation matrix $T \in \mathbb{R}^{1 \times 3}$, where $N_p$ is the number of 2D points in $S$. The Point Lifting Module (PLM) in Figure 2(b) can be formulated as,

$$
\begin{aligned}
P^{3D} &= \text{PLM}(S, E) \\
&= \text{PLM}(P^{2D}, d^+, d^-, T, R_{xyz}, s) \\
&= (\text{Extrude}(P^{2D}, d^+, d^-) \times s)R_{xyz}^{\text{T}} + T \\
&= (\tilde{P}^{3D} \times s)R_{xyz}^{\text{T}} + T
\end{aligned}
\tag{1}
$$

where $\tilde{P}^{3D} \in \mathbb{R}^{2N_p \times 3}$ is in the normalized coordinate system, while $P^{3D} \in \mathbb{R}^{2N_p \times 3}$ is in the global coordinate system, and Extrude converts $P^{2D}$ to $\tilde{P}^{3D}$ along the $z$-axis with distances $d^+$ and $d^-$.

**Dequantization of Argument Tokens.** To facilitate autoregressive modeling, continuous CAD arguments are quantized into discrete tokens. Accordingly, a dequantization step is required when applying the PLM. Instead of hard dequantization via argmax selection, we recover continuous arguments by taking the expectation over the target token. Specifically, given the predicted probability $p$ for the target token and its associated quantized numerical value $v^q$, the dequantized value is defined as,

$$
v^{dq} = p \cdot v^q
\tag{2}
$$

where $v^{dq}$ is the resulting dequantized value used for PLM. In this case, each argument token in the predicted sequence is assigned to its ground-truth counterpart in A3PL.

**Sketch Loss.** The sketch loss is computed on the 2D points predicted in the sketch sequence, as presented in Figure 2(c). Lifting a sketch sequence into 3D via the PLM requires the corresponding extrusion sequence. We provide the ground-truth[1] sequence to ensure that the model accurately captures the true spatial position of the resulting solid. Given a ground-truth CAD sequence $G = \{(S_i, E_i)|i = 1, 2, ..., N_e\}$ composed of $N_e$ sketch sequences $S_{1...N_e}$ and $N_e$ extrusion sequences $E_{1...N_e}$, the corresponding predicted sequence is denoted as $\hat{G} = \{(\hat{S}_i, \hat{E}_i)|i = 1, 2, ..., N_e\}$. The sketch loss is defined as:

$$
\mathcal{L}_{skt} = \frac{1}{N} \sum_{i=1}^{N_e} \left\| \text{PLM}(\hat{S}_i, E_i) - \text{PLM}(S_i, E_i) \right\|_2^2
\tag{3}
$$

where $N = 2\sum_{i=1}^{N_e} N_{p_i}$, and $N_{p_i}$ refers to the number of 2D points in $S_i$.

---

[1] Unless otherwise specified, *given* and *ground-truth* are used interchangeably throughout this paper.

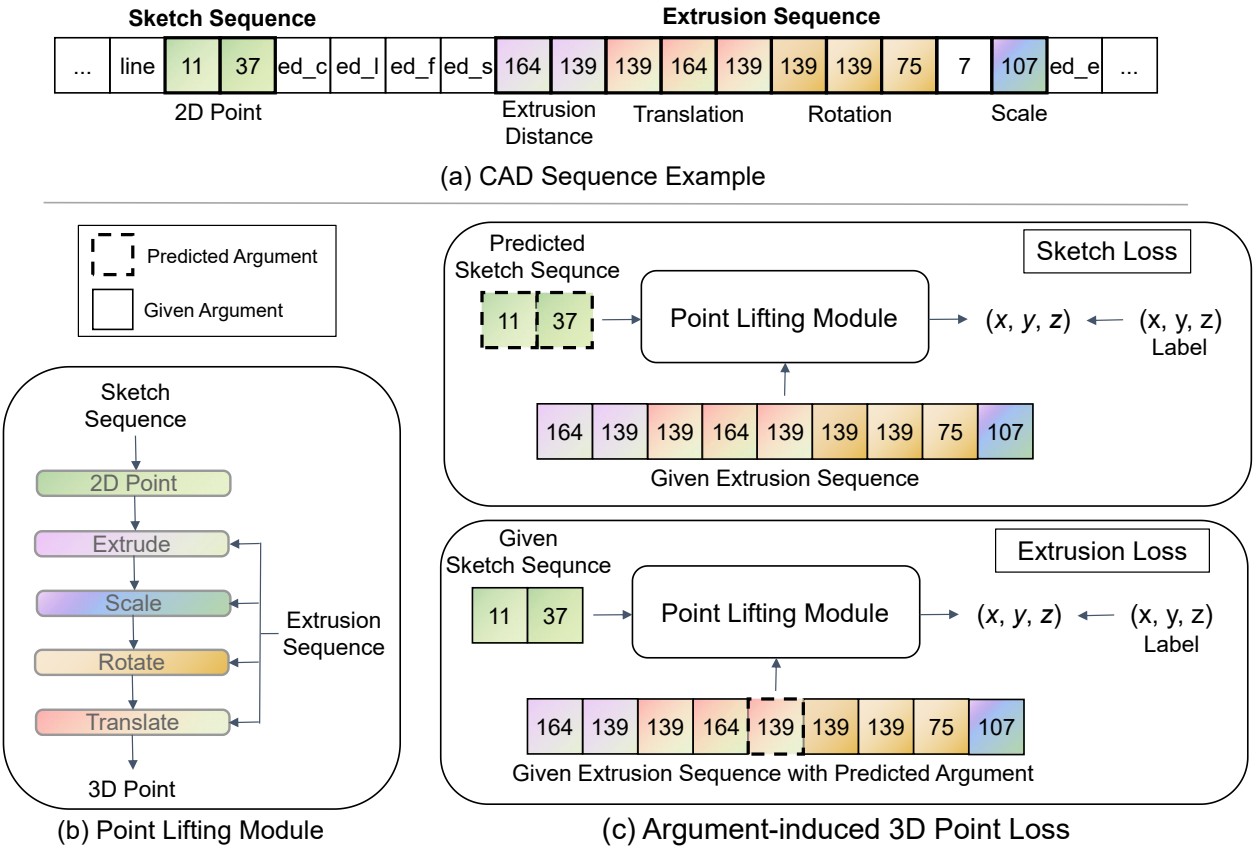

*Figure 2.* Illustration of Argument-induced 3D Point Loss (A3PL). (a) An example of CAD sequence. ed_c, ed_l, ed_f, ed_s. ed_e denote `<end_curve>`, `<end_loop>`, `<end_face>`, `<end_sketch>`, `<end_extrusion>`, respectively. A sketch sequence comprises primitive types, argument tokens representing 2D points, and special end tokens. For simplicity, we use 2D points to denote the sketch sequence in the figure. An extrusion sequence consists of 10 argument tokens followed by a ed_e. (b) Point Lifting Module (PLM) receives a sketch and an extrusion sequence as input and produces 3D points in the global coordinate system. (c) A3PL employs a sketch loss to supervise 2D points in the sketch sequence and an extrusion loss to supervise arguments in the extrusion sequence.

**Extrusion Loss.** Unlike the sketch loss, the extrusion loss is computed when predicting $\hat{E}_i$, where the corresponding sketch sequence $S_i$ is already available as context. Therefore, the loss can be defined based on both $S_i$ and the predicted arguments in $\hat{E}_i$. Since $\hat{E}_i$ is generated in an autoregressive manner, different argument tokens become available progressively during generation. To ensure that the supervision remains consistent with the autoregressive process, PLM is applied to each argument token of $\hat{E}_i$ individually, lifting the 2D points in $S_i$ into 3D according to the currently available extrusion information.

Let $\hat{E}_{i,j}$ denote the sequence obtained from $E_i \in \mathbb{R}^9$ (ignoring the boolean argument token for simplicity) by replacing its $j$-th argument token with the corresponding token from $\hat{E}i$, as illustrated in Figure 2(c), where $j = 1, \ldots, 9$. In this way, $\hat{E}i, j$ preserves the ground-truth extrusion configuration except for the $j$-th argument, enabling the geometric impact of each predicted token to be evaluated independently. Eq. 4 defines the extrusion loss as follows,

$$\mathcal{L}_{ext} = \frac{1}{9N} \sum_{i=1}^{N_e} \sum_{j=1}^{9} \left\| \text{PLM}(S_i, \hat{E}_{i,j}) - \text{PLM}(S_i, E_i) \right\|_2^2$$

(4)

where $N$ and $N_e$ are the same as in Eq. 3.

### 3.3. Extrusion Sequence Augmentation

Extrusion Sequence Augmentation (ESAug) improves the robustness of geometric supervision by reducing reliance on a fixed extrusion context during training, as shown in Figure 3. Using only the ground-truth extrusion sequence in A3PL may cause the model to overfit to specific transformation configurations. To address this, ESAug replaces the ground-truth sequence with a randomly sampled valid sequence with a fixed probability. The sampled parameters are drawn from the same parameter space while satisfying CAD grammar constraints, encouraging the model to learn sketch-to-geometry relationships that are less dependent on

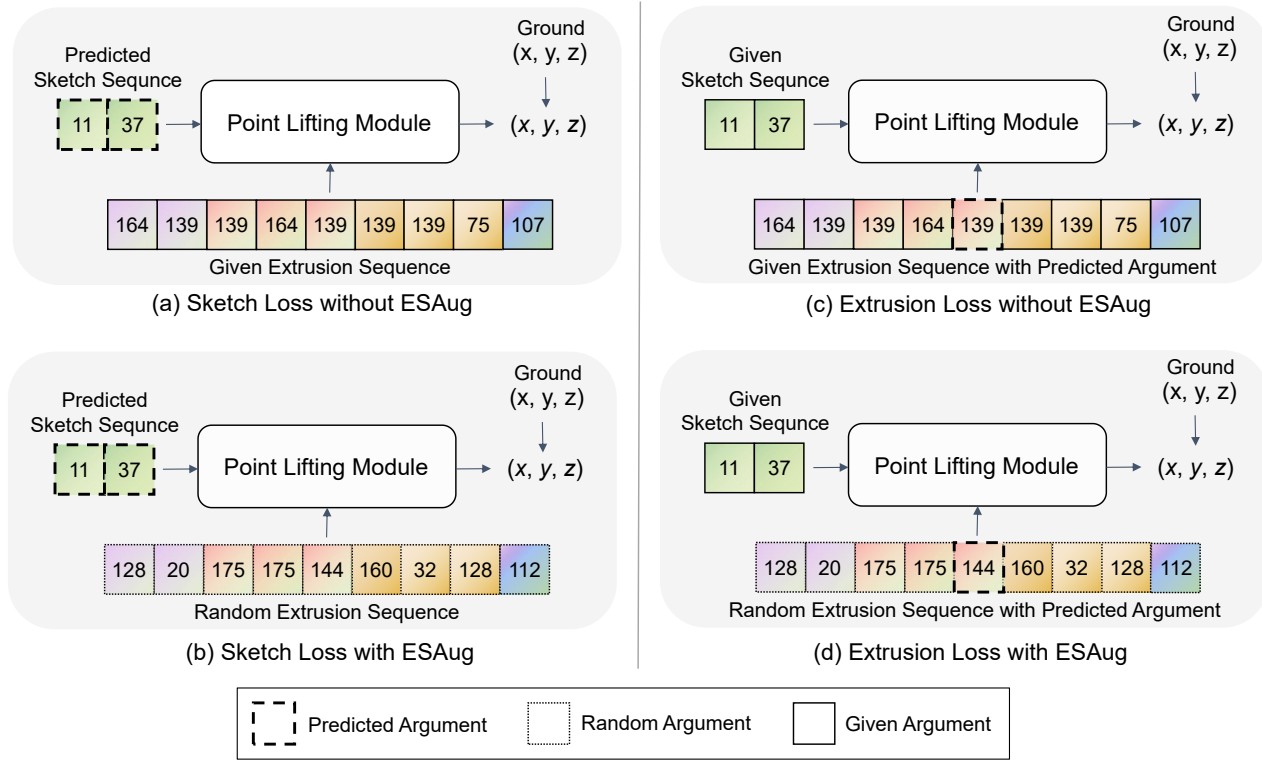

*Figure 3.* Illustration of Extrusion Sequence Augmentation (ESAug). ESAug randomly perturbs the extrusion sequence used for geometric supervision while preserving the autoregressive generation context, encouraging robust learning under varying extrusion configurations.

a specific extrusion configuration.

In our CAD representation, extrusion tokens follow sketch tokens, so under the autoregressive causal mask, only partial extrusion information is visible during sketch generation. ESAug is applied to both sketch and extrusion losses, differing only in the visible context under the causal mask. For sketch loss, future extrusions remain masked, while for extrusion loss, future extrusion parameters are partially masked but the corresponding sketch context is preserved. Randomizing the extrusion sequence in both branches reduces overfitting to a fixed context and promotes a more robust sketch-to-geometry mapping. More details are provided in the supplementary material.

### 3.4. Grammar-constrained Operator

The Grammar-Constrained Operator (GCO) reduces learning complexity and enforces structural validity in CAD sequence generation. As shown in Figure 4, it applies four grammar constraints to structural tokens for primitive and extrusion constructions. During training, structural tokens are masked in the loss computation. During inference, they are deterministically recovered under these constraints and appended to the sequence to condition subsequent autoregressive generation.

**Structural Token.** Structural tokens encode grammar-level elements that determine the high-level structure of CAD construction operations. In contrast, semantic tokens represent executable modeling instructions, including operation types (e.g., primitives and extrusions), their arguments, and control signals for ending loops, faces, or sketches (e.g., `<end_loop>`, `<end_face>`, `<end_sketch>`).

**Constraints.** Grammar constraints define the permissible structure of semantic tokens for different CAD operations to maintain syntactic validity. As in Figure 4, there are four grammar constraints applied.

**Line Constraint:** A line operation begins with `<line>`, followed by 2 arguments, and ends with `<end_curve>`.
**Arc Constraint:** An arc operation begins with `<arc>`, followed by 4 arguments, and ends with `<end_curve>`.
**Circle Constraint:** A circle operation begins with `<circle>`, followed by 4 arguments, and ends with `<end_curve>`.
**Extrusion Constraint:** An extrusion operation is represented by 10 arguments (Figure 2(a)), beginning with `<end_sketch>` and ending with `<end_extrusion>`.

These constraints dictate the number and ordering of structural and semantic tokens for each operation, and are enforced by GCO to maintain structural consistency.

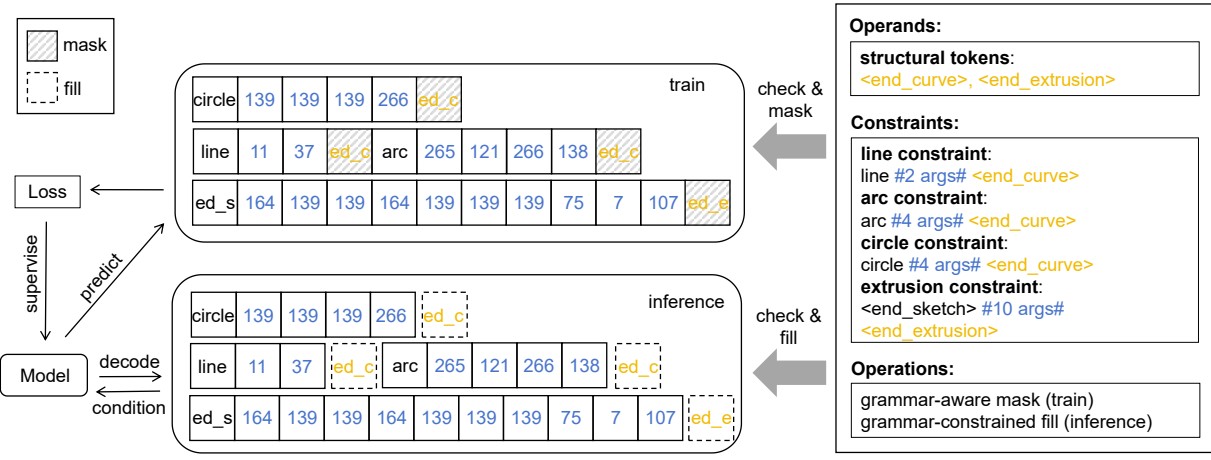

*Figure 4.* Illustration of Grammar-constrained Operator (GCO). ed_c, ed_e, ed_s denote `<end_curve>`, `<end_extrusion>`, `<end_sketch>`, respectively. GCO performs two operations on structural tokens under four grammar constraints. During training, it masks structural tokens when computing the loss. During inference, it fills the structural tokens according to the constraints, and the resulting sequence is then used to generate subsequent tokens.

**GCO Operations.** GCO operates on structural tokens to enforce grammar constraints in CAD sequence generation. It has two operations.

**Grammar-constrained Fill:** During inference, structural tokens are filled according to grammar constraints. Denote the ground-truth sequence and predicted sequence at step $i$ as $S_i = \{t_1, t_2, \ldots, t_i\}$ and $\hat{S}_i = \{\hat{t}_1, \hat{t}_2, \ldots, \hat{t}_i\}$, respectively. The GCO fills structural tokens in $\hat{S}_i$ to produce an intermediate sequence $\tilde{S}_i$, which then conditions the autoregressive generation of the next token:

$$\tilde{S}_i = \text{GCO}(\hat{S}_i), \quad \hat{t}_{i+1} \sim p_\theta(\hat{t} \mid \tilde{S}_i). \quad (5)$$

This enforces that every generated sequence respects the grammar constraints.

**Grammar-aware Mask:** During training, structural tokens are masked in loss computation. Let the ground-truth sequence be $S = \{t_1, t_2, \ldots, t_{N_s}\}$, predicted sequence $\hat{S} = \{\hat{t}_1, \hat{t}_2, \ldots, \hat{t}_{N_s}\}$, and $N_s$ the number of tokens. The masked cross-entropy loss is computed as:

$$\mathcal{L}_{\text{mce}} = - \sum_{t_i \in \text{non-structural}} \log p_\theta(\hat{t} = t_i \mid S_{<i}), \quad (6)$$

where $i = 1, 2, \ldots, N_s$. GCO ensures that the model is not penalized for structural tokens.

### 3.5. Training Objective

We incorporate a masked cross-entropy loss (Eq.6), a sketch loss (Eq.3) and an extrusion loss (Eq.4) for sequence modeling. The final training objective jointly optimizes the sequence-level and token-level objective:

$$\mathcal{L} = \mathcal{L}_{\text{mce}} + \lambda_s \mathcal{L}_{\text{skt}} + \lambda_e \mathcal{L}_{\text{ext}}, \quad (7)$$

where the scalar coefficients $\lambda_s$ and $\lambda_e$ control the relative importance of $\mathcal{L}_{\text{skt}}$ and $\mathcal{L}_{\text{ext}}$, respectively.

## 4. Experiments

### 4.1. Setups

**Dataset.** We conduct experiments on five CAD sequence generation tasks with different input modalities. All datasets are derived from DeepCAD dataset (Wu et al., 2021), following the standard preprocessing protocol as mentioned in previous works (Wu et al., 2021; Khan et al., 2024b; Wang et al., 2025a; Qin et al., 2025; Khan et al., 2024a). Further details are provided in the supplementary material.

**Metrics.** We use F1 scores (Khan et al., 2024b) for sketch primitives and extrusion operations to evaluate the prediction of CAD sequences at the token level. Chamfer Distance (CD) (Khan et al., 2024b) is adopted to measure geometric discrepancies between rendered 3D objects and ground-truth, evaluating whether A3PL effectively associates argument tokens with their underlying 3D geometry. Besides, Invalidity Ratio (IR) is reported to assess the effectiveness of GCO in improving the overall validity of generated CAD sequences.

**Baselines.** To evaluate the effectiveness and generalization of the proposed method, we consider five CAD sequence generation tasks as baselines. Specifically, Text2CAD (Khan et al., 2024b) and CADFusion (Wang et al., 2025a) address text-to-CAD generation, where the former is trained from scratch, whereas the latter leverages a pretrained language model. DeepCAD (Wu et al., 2021) models CAD sequences with an encoder–decoder architecture. Draw-

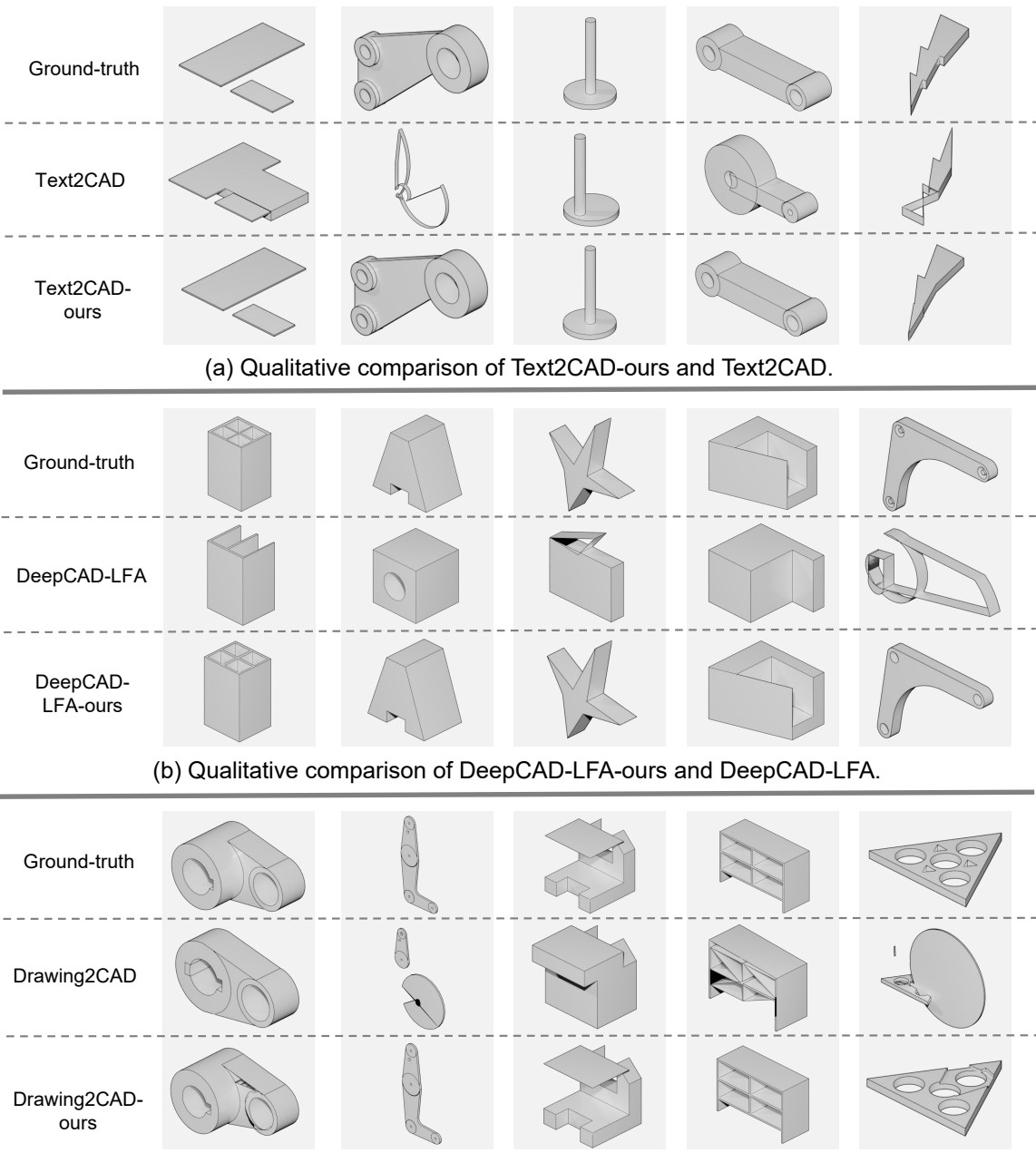

(a) Qualitative comparison of Text2CAD-ours and Text2CAD.

(b) Qualitative comparison of DeepCAD-LFA-ours and DeepCAD-LFA.

(c) Qualitative comparison of Drawing2CAD-ours and Drawing2CAD.

*Figure 5.* Qualitative comparison between our methods and their corresponding baseline counterparts on (a) Text2CAD, (b) DeepCAD-LFA, and (c) Drawing2CAD. Our method better captures the spatial relationships among different solids within an object and more accurately predicts arcs and circles in sketches. However, baseline models often fail to infer appropriate arguments for primitives and extrusion, leading to fragmented geometries or incorrect relative spatial arrangements.

ing2CAD (Qin et al., 2025) takes SVG sketches as input, and DeepCAD-LFA (Wu et al., 2021; Khan et al., 2024a) focuses on point clouds.

**Implementation Details.** All experiments are conducted on 8 NVIDIA RTX 3090 GPUs. For the five experiments in Section 4.2, we adhere to the implementation settings reported in the respective original works to ensure fair comparison. The scalar coefficients $\lambda_s$ and $\lambda_e$ (Secton 3.5) are both 10. We set the probability of ESAug to 0.7 for Text2CAD (Khan et al., 2024b) and Drawing2CAD (Qin et al., 2025), 0.5 for DeepCAD-LFA (Wu et al., 2021; Khan et al., 2024a), and 0.1 for CADFusion (Wang et al., 2025a) and DeepCAD (Wu et al., 2021). Further details are pro-

*Table 1.* Quantitative results of our methods on five different tasks. Performance is evaluated using F1 scores for sketch primitives and extrusion, mean and median Chamfer Distance (CD), and Invalidity Ratio (IR). ↑ and ↓ indicate that higher and lower values are better, respectively. Owing to the differences in CAD representations, F1 scores are not directly comparable and thus cannot be computed for DeepCAD and Drawing2CAD. CD is multiplied by $10^3$.

| Method | F1↑ | | | | CD↓ | | IR↓ |
|---|---|---|---|---|---|---|---|
| | Line | Arc | Circle | Extrusion | Median | Mean | |
| DeepCAD | - | - | - | - | 0.78 | 7.16 | 2.72 |
| DeepCAD-ours | 95.25 | 75.90 | 94.93 | 99.98 | **0.18** | **6.82** | **1.39** |
| Text2CAD | 84.66 | 44.86 | 79.94 | 94.35 | 0.30 | 30.50 | 3.57 |
| Text2CAD-ours | **85.97** | **55.91** | **82.62** | **94.95** | **0.28** | **22.78** | **1.96** |
| CADFusion | 81.63 | 78.92 | 89.42 | 93.45 | 1.73 | 20.86 | 6.02 |
| CADFusion-ours | **82.15** | **82.43** | **91.15** | **93.81** | **1.28** | **17.62** | **1.92** |
| Drawing2CAD | - | - | - | - | 8.81 | 108.80 | 20.31 |
| Drawing2CAD-ours | 76.96 | 39.80 | 68.29 | 89.04 | **0.86** | **21.59** | **1.52** |
| DeepCAD-LFA | 71.03 | 9.78 | 57.96 | 89.77 | 1.82 | 27.19 | 2.14 |
| DeepCAD-LFA-ours | **72.23** | **19.37** | **60.69** | **90.06** | **1.02** | **21.63** | **0.86** |

*Table 2.* CD Mean and IR results on test samples of varying difficulty levels: E (easy), M (medium), and H (hard). Methods annotated with a superscript † denote the corresponding versions of our method listed in Table 1.

| Method | CD Mean↓ | | | IR↓ | | |
|---|---|---|---|---|---|---|
| | E | M | H | E | M | H |
| DeepCAD | 2.56 | 17.3 | 39.7 | 0.20 | 3.12 | 14.2 |
| DeepCAD† | **2.10** | **16.9** | **38.7** | **0.13** | **2.08** | **10.3** |
| Text2CAD | 12.8 | 106. | 73.2 | 1.23 | 8.22 | 16.9 |
| Text2CAD† | **11.8** | **47.8** | **71.5** | **0.50** | **4.32** | **9.75** |
| CADFuson | 10.8 | 29.6 | 37.8 | 2.93 | 11.2 | 18.3 |
| CADFusion† | **10.3** | **27.4** | **31.2** | **0.74** | **3.18** | **7.25** |
| Drawing2CAD | 64.5 | 124. | 184 | 11.2 | 32.2 | 48.1 |
| Drawing2CAD† | **18.3** | **31.4** | **33.2** | **0.86** | **3.01** | **4.86** |
| DeepCAD-LFA | 25.0 | 33.4 | 40.0 | 0.58 | 2.08 | 3.02 |
| DeepCAD-LFA† | **18.6** | **26.3** | **36.2** | **0.48** | **1.76** | **2.01** |

vided in the supplementary material.

### 4.2. Evaluation

**Qualitative Results.** Figure 5 presents qualitative results of three baseline models, Text2CAD, DeepCAD-LFA, and Drawing2CAD, which take text, point clouds, and SVG sketches as input modalities, respectively. For objects composed of more than two solids, these baseline methods often fail to infer appropriate arguments for primitives and extru-

sion, leading to fragmented geometries or incorrect relative spatial arrangements. In contrast, our model more effectively captures 3D geometric relationships, allowing independent solids to be positioned based on input-specified layouts. Although minor components are occasionally omitted, the proposed method reliably preserves the overall structure by accurately modeling the shapes and spatial configurations of the major parts. Moreover, our model demonstrates superior performance on complex geometries involving arcs and circles, which is consistent with the higher F1 scores reported for these two primitive types in Table 1.

**Quantitative Results.** Table 1 summarizes the quantitative comparison with baseline methods. Our model achieves higher F1 scores on arcs and circles, with particularly notable improvements on arcs. Specifically, it yields a 25% gain over Text2CAD and a 100% gain over DeepCAD-LFA, indicating its superior ability to capture complex sketch geometries. In addition, our model attains comparable or better performance in terms of median and mean CD. It reduces the mean CD by approximately 25% on Text2CAD and by up to a factor of five on Drawing2CAD. These results demonstrate the effectiveness of A3PL in providing geometric supervision for individual argument tokens, thereby facilitating the learning of accurate spatial relationships. Moreover, our model consistently outperforms all baseline methods on IR, highlighting the role of GCO in reducing learning complexity and improving sequence validity. We further divide the DeepCAD dataset into three difficulty levels. As shown in Table 2, for Medium and Hard levels, our methods consistently improve the mean CD across multiple tasks and significantly outperform most baselines in IR, particularly on Text2CAD and Drawing2CAD. Further

*Table 3.* Ablation study of different model variants. $M_{baseline}$ denotes the baseline model. $M_{A3PL}$ uses only A3PL without ESAug. $M_{A3PL+ESAug}$ applies A3PL together with ESAug. $M_{GCO}$ adds GCO alone. $M_{A3PL+ESAug+GCO}$ incorporates all proposed methods. The results for the five CAD generation models {DeepCAD, Text2CAD, CADFusion, Drawing2CAD, DeepCAD-LFA} are reported in a fixed order within each cell: DeepCAD / Text2CAD / CADFusion / Drawing2CAD / DeepCAD-LFA. The probability of ESAug follows the same setting as in Table 1.

| Method | F1 Sketch↑ | CD Mean↓ | IR↓ |
|---|---|---|---|
| $M_{baseline}$ | - /69.8/83.3/ - /46.3 | 7.1 /30.5/20.8/108./27.1 | 2.7/3.5/6.0/20./2.1 |
| $M_{A3PL}$ | 86.1/69.9/82.9/60.8/49.0 | 8.9 /25.2/18.9/23.7/24.4 | 1.9/2.7/3.7/2.6/1.5 |
| $M_{A3PL+ESAug}$ | 86.7/70.2/84.0/61.0/49.1 | 7.3 /23.9/18.6/22.2/23.0 | 1.6/2.5/3.5/2.2/1.1 |
| $M_{GCO}$ | 88.2/71.5/84.8/**61.7**/50.1 | 9.6 /26.0/21.4/26.2/26.1 | 1.4/2.2/2.5/1.9/0.9 |
| $M_{A3PL+ESAug+GCO}$ | **88.7**/**74.8**/**85.0**/**61.7**/**50.8** | **6.8** /**22.7**/**17.6**/**21.5**/**21.6** | **1.3**/**1.9**/**1.9**/**1.5**/**0.8** |

details are included in the supplementary material.

### 4.3. Ablation Study

We conduct ablation studies on the five baseline models to examine the effectiveness of A3PL in conjunction with ESAug, and also the impact of GCO. In Table 3, the *baseline* variant corresponds to the original methods. The *A3PL* variant applies A3PL without ESAug, whereas the *A3PL+ESAug* variant introduces ESAug on top of A3PL. The *GCO* variant adopts only GCO. The *A3PL+ESAug+GCO* variant combines A3PL with ESAug and GCO within the baseline models. The probability of ESAug follows Section 4.2. More details are provided in the supplementary material.

**Geometric Supervision.** As shown in Table 3, the *A3PL* variants achieve performance comparable to the *baseline* variants on F1 Sketch. However, the *A3PL* variants outperform the *baseline* variants on CD Mean, indicating that A3PL provides effective geometric feedback. The *A3PL+ESAug* variants further improve upon the *A3PL* variants across all five tasks on F1 Sketch and CD Mean, demonstrating the benefit of ESAug for learning 3D geometry. Compared with the *baseline* variants, the *GCO* variants show higher F1 Sketch and lower CD Mean, suggesting that improved accuracy on primitive type leads to better overall geometric prediction.

**Sequence Validity.** As shown in Table 3, the *A3PL+ESAug* variants improve IR over the *baseline* variants, suggesting that geometry-aware augmentation provides additional regularization for valid sequence generation. The *GCO* variants further deliver consistent gains on F1 Sketch and IR across all tasks, demonstrating that enforcing grammar constraints directly reduces learning complexity and prevents structurally invalid sequences.

**Combination of *A3PL+ESAug* and *GCO*.** As shown in Table 3, the *A3PL+ESAug+GCO* variants consistently outperform both the *A3PL+ESAug* and *GCO* variants across all metrics. This improvement stems from the complementary mechanisms of the two components: A3PL with ESAug provides token-wise geometric feedback, while GCO constrains structural predictions and reduces the effective learning space. Their combination therefore yields more accurate geometry prediction and higher sequence validity than either method alone.

## 5. Conclusion

For CAD sequence generation, we propose A3PL combined with ESAug to provide token-wise 3D geometric supervision, and introduce GCO to enforce the validity of generation. A3PL enables the model to learn geometric consistency at the token level, while GCO ensures that the output programs adhere to grammar constraints. Experimental results across multiple modalities demonstrate that our approach improves CAD program learning in terms of both accuracy and validity. In future work, we plan to extend geometric supervision to non-numeric tokens and investigate strategies to enhance the compositional generalization.

## Acknowledgements

We sincerely thank the anonymous reviewers for their constructive comments and thoughtful suggestions, which have significantly improved this work. We also thank our colleagues and collaborators for valuable discussions to the development of the ideas presented in this paper.

## Impact Statement

This work aims to advance CAD sequence modeling by improving geometric supervision and sequence validity. The potential societal and ethical implications are similar to those commonly associated with advances in machine learning for generative tasks, and we do not identify any specific concerns unique to this work. Overall, this paper presents research intended to advance the field of machine learning without foreseeable broader negative societal impacts beyond those typical for this area.

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

# A. CAD Representation

We adopt the CAD representation from Text2CAD (Khan et al., 2024b), with minor extensions to include three tokens indicating primitive types. The resulting grammar-based parameterization of primitives is defined as follows:

1. **Line:** $<$ Line $>$, start coordinate, end coordinate, $<$ end_curve $>$.

2. **Arc:** $<$ Arc $>$, start coordinate, middle coordinate, end coordinate, $<$ end_curve $>$.

3. **Circle:** $<$ Circle $>$, center coordinate, top-most coordinate, $<$ end_curve $>$.

As in Figure 2(a), each extrusion sequence consists of 10 arguments with a <end_extrusion>. These arguments are listed in the same order as they appear in the sequence.

1. **Extrusion Distance:** Two arguments, $d^+$ and $d^-$, define extrusion distances in the positive and negative normal directions of the sketch plane.

*Table 4.* CAD sequence representation in Section 3.1.

| Sequence Type | Token Type | Token Value | Token Representation | Description |
|---|---|---|---|---|
| | $pad$ | 0 | $(0,0)$ | Padding Token |
| | $start$ | 1 | $(1,0)$ | Start Token |
| | $end$ | 1 | $(1,0)$ | End Token |
| Sketch Sequence | $e_s$ | 2 | $(2,0)$ | End Sketch |
| | $e_f$ | 3 | $(3,0)$ | End Face |
| | $e_l$ | 4 | $(4,0)$ | End Loop |
| | $e_c$ | 5 | $(5,0)$ | End Curve |
| | $line$ | 6 | $(6,0)$ | Line Type |
| | $arc$ | 7 | $(7,0)$ | Arc Type |
| | $circle$ | 8 | $(8,0)$ | Circle Type |
| | $(p_x, p_y)$ | $[14, 269]^2$ | $(p_x, p_y)$ | Coordinates |
| Extrusion Sequence | $d^+$ | $[14, 269]$ | $(d^+, 0)$ | Extrusion Distance Towards Sketch Plane Normal |
| | $d^-$ | $[14, 269]$ | $(d^-, 0)$ | Extrusion Distance Opposite Sketch Plane Normal |
| | $\tau_x$ | $[14, 269]$ | $(\tau_x, 0)$ | Sketch Plane Origin |
| | $\tau_y$ | $[14, 269]$ | $(\tau_y, 0)$ | |
| | $\tau_z$ | $[14, 269]$ | $(\tau_z, 0)$ | |
| | $\theta$ | $[14, 269]$ | $(\theta, 0)$ | Sketch Plane Orientation |
| | $\phi$ | $[14, 269]$ | $(\phi, 0)$ | |
| | $\gamma$ | $[14, 269]$ | $(\gamma, 0)$ | |
| | $\sigma$ | $[14, 269]$ | $(\sigma, 0)$ | Sketch Scaling Factor |
| | $\beta$ | $\{10, 11, 12, 13\}$ | $(\beta, 0)$ | Boolean Operation = {New, Cut, Join, Intersect) |
| | $e_e$ | 9 | $(9,0)$ | End Extrusion |

2. **Translation:** Three arguments, $\tau_x$, $\tau_y$ and $\tau_z$, specify the translation of the sketch plane.

3. **Rotation:** Three parameters, $\theta$, $\phi$ and $\gamma$, determine the orientation of the sketch plane.

4. **Boolean Operation:** One argument, $\beta$, specifies the extrusion operation, with four possible types: new solid, cut, join, and intersection.

5. **Scale:** One argument, $\sigma$, controls the scaling of the 2D sketches.

Except for the boolean operation and end tokens, all 2D sketch and extrusion parameters are quantized to 8 bits. The end coordinates of lines and arcs are not explicitly represented and are predicted by the following primitive or the first primitive of the loop. Details are provided in Table 4.

## B. Discussion

Although our method is instantiated on the sketch–extrude CAD representation, the underlying assumptions of the proposed A3PL are not specific to this particular representation. Fundamentally, most programmatic CAD representations, whether expressed as sketch–extrude sequences, Python-based CAD scripts, or other domain-specific languages, share a common structure: they require the model to predict sketch-related arguments and extrusion-related arguments to determine the final 3D geometry. The primary difference among these representations lies in the choice of programming language and syntactic abstraction, rather than in the underlying geometric semantics. As a result, A3PL, which supervises argument tokens through their induced 3D point geometry, can be naturally and effectively extended to other CAD representations. As long as the predicted arguments can be mapped to geometric transformations, token-wise geometric supervision remains applicable.

Similarly, GCO is not inherently tied to the sketch–extrude representation. Its primary goal is to reduce learning complexity and improve the validity of generated CAD programs by handling structural tokens that are deterministically defined by the underlying grammar. This principle is independent of the specific CAD language. In practice, GCO can be adapted to other CAD representations by identifying the corresponding set of structural tokens that are uniquely determined by grammar rules. These structural components can often be derived automatically from the formal grammar, enabling straightforward extension of GCO beyond sketch–extrude workflows. Moreover, as CAD representations evolve to include more complex and diverse semantic operations, GCO continues to operate at the syntactic level of the program. By focusing exclusively on structural tokens defined by the grammar, GCO remains largely decoupled from the complexity of semantic operators, allowing it to maintain its role in improving sequence validity and simplifying the learning process.

## C. Extrusion Sequence Augmentation

Here, we provide a theoretical justification demonstrating that ESAug improves the model's generalization ability.

1. **Preliminaries.** The PLM defines a family of extrusion-induced transformations:

$$\mathcal{T} = \{T_E : E \in \mathcal{E}_{valid}\}, \tag{8}$$

where $T_E(\cdot)$ denotes the transformation that maps a 2D sketch into 3D space under extrusion $E$, and $\mathcal{E}_{valid}$ denotes the set of valid extrusion sequences that produce geometrically feasible 3D shapes.

2. **Ambiguity under ground-truth extrusion.** Let $S$ denote the ground-truth sketch and $\hat{S}$ the predicted sketch. The sketch loss without ESAug is defined as:

$$L_{E_{gt}}(\hat{S}) = \|T_{E_{gt}}(\hat{S}) - T_{E_{gt}}(S)\|^2, \tag{9}$$

Under the ground-truth extrusion $E_{gt}$, this constraint is not sufficient to uniquely determine $\hat{S}$. In particular, there may exist $\hat{S} \neq S$ such that:

$$T_{E_{gt}}(\hat{S}) = T_{E_{gt}}(S), \tag{10}$$

leading to spurious solutions that only align in a specific coordinate system.

3. **ESAug removes spurious solutions.** With ESAug, the loss becomes:

$$L_{\text{ESAug}}(\hat{S}) = \mathbb{E}_{E \sim p_{\text{aug}}}\big[\|T_E(\hat{S}) - T_E(S)\|^2\big], \tag{11}$$

where $p_{\text{aug}}$ denotes the augmentation-induced distribution over valid extrusion parameters. If $\mathcal{T} = \{T_E\}$ is sufficiently rich, then for any $\hat{S} \neq S$, there exists at least one $E \in \mathcal{E}_{valid}$ such that:

$$T_E(\hat{S}) \neq T_E(S). \tag{12}$$

In practice, when $p_{\text{aug}}$ sufficiently covers $\mathcal{E}_{valid}$, this implies:

$$L_{\text{ESAug}}(\hat{S}) > 0 \quad \forall \hat{S} \neq S \ \ (\text{approximately}), \tag{13}$$

which is a standard approximation when Monte Carlo sampling is used to estimate expectations over $\mathcal{T}$. ESAug suppresses spurious minima and encourages the solution $\hat{S}$ to be consistent across a broad set of extrusion-induced transformations. This effectively constrains the hypothesis space to transformation-consistent solutions.

4. **ESAug improves generalization.** In this analysis, we employ the Empirical Risk Minimization (ERM) framework (Bartlett & Mendelson, 2002). At test time, the extrusion sequences may differ from those seen during training. The test risk can be written as:

$$L_{\text{test}}(\hat{S}) = \mathbb{E}_{E \sim p_{\text{test}}}\big[\|T_E(\hat{S}) - T_E(S)\|^2\big]. \tag{14}$$

By adding and subtracting the training expectation, we obtain:

$$L_{\text{test}}(\hat{S}) = L_{\text{ESAug}}(\hat{S}) + \Big(\mathbb{E}_{E \sim p_{\text{test}}}[\ell(\hat{S}, E)] - \mathbb{E}_{E' \sim p_{\text{aug}}}[\ell(\hat{S}, E')]\Big), \tag{15}$$

where $\ell(\hat{S}, E) = \|T_E(\hat{S}) - T_E(S)\|^2$. Supposing $p_{\text{test}} = \mathcal{E}_{valid}$, this leads to the bound:

$$L_{\text{test}}(\hat{S}) \leq L_{\text{ESAug}}(\hat{S}) + \sup \Big|\mathbb{E}_{E \sim \mathcal{E}_{valid}}[\ell(\hat{S}, E)] - \mathbb{E}_{E' \sim p_{\text{aug}}}[\ell(\hat{S}, E')]\Big|. \tag{16}$$

The second term reflects the distribution mismatch between training and test extrusion sequences. By sampling $E'$ from a broad valid space, ESAug enlarges the support of $p_{\text{aug}}$, thereby reducing this mismatch term. As a result, minimizing $L_{\text{ESAug}}$ provides better control over $L_{\text{test}}$, leading to improved generalization.

# D. Implementation Details

This section provides additional implementation details for all experiments reported in the main paper. Our goal is to clarify experiment-specific configurations that cannot be fully described in the main text due to space limitations. All experiments are conducted using the same overall training framework and hardware setup, while closely following the original implementation settings of each baseline to ensure fair comparison.

In the following paragraphs, we describe the detailed training configurations for each of the five CAD sequence generation tasks considered in our experiments, namely Text2CAD (Khan et al., 2024b), CADFusion (Wang et al., 2025a), DeepCAD (Wu et al., 2021), Drawing2CAD (Qin et al., 2025), and DeepCAD-LFA (Khan et al., 2024a; Wu et al., 2021). Unless otherwise specified, hyperparameters not explicitly mentioned are kept consistent with those reported in the corresponding original works.

**DeepCAD-ours.** The model is trained using the Adam optimizer (Kinga et al., 2015) with a learning rate of 0.001 and a linear warm-up over the first 2,000 steps. A dropout rate of 0.1 is applied to all Transformer layers, and gradients are clipped to a maximum norm of 1.0 during backpropagation. Training is conducted for 1,000 epochs with a batch size of 512. The scalar coefficients $\lambda_s$ and $\lambda_e$ are both set to 10, and the ESAug probability is set to 0.1. The decoder is trained using an autoregressive teacher-forcing (Williams & Zipser, 1989) strategy. DeepCAD-ours adopts the same model architecture and hyperparameter settings as DeepCAD.

**Text2CAD-ours.** The model is trained using the AdamW optimizer (Loshchilov & Hutter, 2017) with a learning rate of 0.001. A dropout rate of 0.1 is applied, and the maximum sequence length is fixed to 512 tokens. Training is performed with a batch size of 512 for 160 epochs. The scalar coefficients $\lambda_s$ and $\lambda_e$ are both set to 10, and the ESAug probability is

set to 0.7. Consistent with the findings in CADFusion (Wang et al., 2025a), we observe a clear performance gap between abstract-level and expert-level prompts. As a result, we restrict our experiments to expert-level prompts for both Text2CAD and Text2CAD-ours.

**CADFusion-ours.** LLaMA-3-8B (AI@Meta, 2024) is used as the LLM backbone with a maximum token length of 1024. Efficient fine-tuning is performed via Low-Rank Adaptation (LoRA) (Hu et al., 2022) with rank $r = 32$ and scaling factor $\alpha = 32$. The initial sequential learning stage is trained for 40 epochs using the AdamW optimizer with a learning rate of $1 \times 10^{-4}$. Five cycles of 5-epoch visual feedback and 1-epoch sequential learning are performed on the preference data. The batch size is set to 8. The scalar coefficients $\lambda_s$ and $\lambda_e$ are both set to 10, and the ESAug probability is set to 0.1.

**Drawing2CAD-ours.** Training is conducted for 200 epochs with a batch size of 512. We optimize the model using Adam with a learning rate of 0.001, and employ a linear warm-up schedule for the initial 2000 steps. A dropout rate of 0.1 is applied across all Transformer blocks, and gradient clipping is performed with a maximum norm of 1.0. The scalar weights $\lambda_s$ and $\lambda_e$ are both set to 10, and the ESAug probability is set to 0.7. The decoder is trained with an autoregressive teacher-forcing scheme.

**DeepCAD-LFA-ours.** We follow the experimental settings of DeepCAD (Wu et al., 2021) and CAD-SIGNet (Khan et al., 2024a). An LFA encoder (Hu et al., 2021) is employed to learn geometric representations from point clouds. The number of nearest neighbors in KNN is set to 16, and the decimation ratio is fixed at 4. We use four LFA layers with output dimensions $\{32, 128, 256, 512\}$. Each sample contains 8192 randomly sampled points. For the decoder, we adopt eight MultiModalTransformer layers as in Text2CAD. The first two layers apply self-attention only, while the remaining layers perform cross-attention over the token representations produced by the LFA encoder. The decoder hidden dimension is set to 512. The network is trained for 150 epochs with a batch size of 128. Curriculum learning (Bengio et al., 2009) is applied during the first 15 epochs by progressively sorting CAD sequences according to the number of curves. Training is performed using the AdamW optimizer with a learning rate of 0.001, along with an ExponentialLR scheduler with a decay factor of $\gamma = 0.999$. The dropout rate is set to 0.1. The scalar weights $\lambda_s$ and $\lambda_e$ are both set to 10, and the ESAug probability is set to 0.5.

# E. Baseline

In Table 1, we reproduce the results of DeepCAD, CADFusion, and Drawing2CAD, obtaining performance comparable to that reported in the original works. For Text2CAD, we follow CADFusion and use only expert-level prompts, yielding results consistent with those reported in the paper. For DeepCAD-LFA, results are produced using the same experimental settings as DeepCAD-LFA-ours.

# F. Ablation Study

Table 5 presents more detailed ablation study results. Beyond overall sketch accuracy, we further analyze performance on curved primitives (Arc and Circle) and Median CD, which more directly reflect the ability to capture fine-grained geometric structure. As shown in Table 5, the $GCO$ variants consistently improve Arc and Circle F1 scores over the $baseline$ across all five models, indicating that grammar constraints is particularly beneficial for learning curvature primitives.

Notably, on Text2CAD, the Arc F1 score increases from 44.86 ($baseline$) to 49.99 ($GCO$) and further to 55.91 ($ours$), accompanied by a substantial reduction in Median CD (from 0.30 to 0.28). This suggests that geometric feedback and grammar constraints effectively mitigate ambiguity in curved primitive prediction, which is common in purely sequence-driven models.

Similarly, for DeepCAD-LFA, which integrates point cloud perception, DeepCAD-LFA$_{A3PL}$ already yields noticeable improvements on Arc accuracy, while DeepCAD-LFA$_{ESAug}$ further reduces the Median CD from 1.21 to 1.04. DeepCAD-LFA$_{ours}$ achieves the lowest Median CD of 1.02, demonstrating that geometric supervision improves not only primitive recognition but also the consistency of reconstructed geometry at the instance level.

For Drawing2CAD, which exhibits large geometric variance in the baseline, the introduction of ESAug significantly reduces Median CD from 8.81 to below 1.0, while steadily improving Arc and Circle F1 scores. This highlights the robustness of ESAug in stabilizing geometric learning under noisy or ambiguous sketch inputs.

Across all experiments, the $ours$ variants consistently achieve the strongest performance on curved primitives and Median

*Table 5.* Detailed ablation study of different model variants across five CAD generation frameworks. For each method, we report fine-grained F1 scores for different primitive types (Line, Arc, Circle, and Extrusion), as well as Chamfer Distance (CD) and Invalid Rate (IR). Results are reported separately for DeepCAD, Text2CAD, CADFusion, Drawing2CAD, and DeepCAD-LFA.

| Method | F1↑ | | | | CD↓ | | IR↓ |
| --- | --- | --- | --- | --- | --- | --- | --- |
| | Line | Arc | Circle | Extrusion | Median | Mean | |
| DeepCAD$_{baseline}$ | - | - | - | - | 0.78 | 7.16 | 2.72 |
| DeepCAD$_{A3PL}$ | 94.73 | 69.65 | 93.96 | 99.95 | 0.19 | 8.95 | 1.94 |
| DeepCAD$_{ESAug}$ | 94.68 | 71.22 | 94.21 | 99.95 | 0.21 | 7.32 | 1.62 |
| DeepCAD$_{GCO}$ | 94.86 | 74.88 | 94.74 | 99.97 | 0.19 | 9.66 | 1.49 |
| DeepCAD$_{ours}$ | **95.25** | **75.9** | **94.93** | **99.98** | **0.18** | **6.82** | **1.39** |
| Text2CAD$_{baseline}$ | 84.66 | 44.86 | 79.94 | 94.35 | 0.3 | 30.5 | 3.57 |
| Text2CAD$_{A3PL}$ | 84.29 | 46.12 | 79.35 | 94.37 | 0.33 | 25.23 | 2.78 |
| Text2CAD$_{ESAug}$ | 84.38 | 47.1 | 79.24 | 93.43 | 0.33 | 23.99 | 2.58 |
| Text2CAD$_{GCO}$ | 84.26 | 49.99 | 80.1 | 94.37 | 0.32 | 26.08 | 2.29 |
| Text2CAD$_{ours}$ | **85.97** | **55.91** | **82.62** | **94.95** | **0.28** | **22.78** | **1.96** |
| CADFusion$_{baseline}$ | 81.63 | 78.92 | 89.42 | 93.45 | 1.73 | 20.86 | 6.02 |
| CADFusion$_{A3PL}$ | 81.45 | 77.92 | 89.36 | 92.28 | 1.64 | 18.92 | 3.76 |
| CADFusion$_{ESAug}$ | 81.55 | 80.24 | 90.11 | 93.14 | 1.58 | 18.65 | 3.58 |
| CADFusion$_{GCO}$ | 81.48 | 82.43 | 90.52 | 92.98 | 2.21 | 21.45 | 2.52 |
| CADFusion$_{ours}$ | **82.15** | **82.85** | **91.15** | **93.81** | **1.28** | **17.62** | **1.92** |
| Drawing2CAD$_{baseline}$ | - | - | - | - | 8.81 | 108.8 | 20.31 |
| Drawing2CAD$_{A3PL}$ | 75.21 | 39.21 | 67.98 | 89.04 | 0.86 | 23.72 | 2.6 |
| Drawing2CAD$_{ESAug}$ | 75.33 | 39.25 | 68.34 | 89.45 | 0.84 | 22.26 | 2.28 |
| Drawing2CAD$_{GCO}$ | 74.66 | 39.8 | 68.29 | 89.42 | 0.98 | 26.25 | 1.9 |
| Drawing2CAD$_{ours}$ | **76.96** | **40.68** | **69.79** | **89.85** | **0.81** | **21.59** | **1.52** |
| DeepCAD-LFA$_{baseline}$ | 71.03 | 9.78 | 57.96 | 89.77 | 1.82 | 27.19 | 2.14 |
| DeepCAD-LFA$_{A3PL}$ | 72.25 | 16.8 | 57.95 | 89.86 | 1.21 | 24.46 | 1.59 |
| DeepCAD-LFA$_{ESAug}$ | 72.23 | 16.83 | 58.07 | 89.97 | 1.04 | 23.06 | 1.19 |
| DeepCAD-LFA$_{GCO}$ | 72.11 | 17.08 | 60.69 | 89.34 | 1.12 | 26.11 | 0.94 |
| DeepCAD-LFA$_{ours}$ | **72.29** | **19.37** | **61.14** | **90.06** | **1.02** | **21.63** | **0.86** |

CD, confirming that token-wise geometric supervision and grammar constraints are complementary. Their combination yields more accurate curvature modeling and more stable geometric reconstruction than either component alone.

We conducted experiments on GeoCAD (Zhang et al., 2026). We trained GeoCAD and GeoCAD-ours with the code released. The training pipeline of GeoCAD consists of two stages. We are able to perform the Stage II training. The corresponding results are presented in Table 6.

# G. Dataset Complexity Levels

In Table 2, we categorize the dataset into three complexity levels—easy, medium, and hard—based on the number of curves in each CAD sequence. The number of curves ranges from 1 to approximately 50. Accordingly, sequences with 1–10 curves are labeled as easy, those with 11–20 curves as medium, and those with more than 20 curves as hard. This partition yields approximately 5.8k easy, 1.5k medium, and 0.7k hard samples in the test set.

*Table 6.* Quantitative results of our methods on GeoCAD (Zhang et al., 2026). ↑ and ↓ indicate that higher and lower values are better, respectively. The metrics are consistent with GeoCAD.

| Method | CD Mean↓ | COV↑ | MMD↓ | JSD↓ | IR↓ |
|---|---|---|---|---|---|
| GeoCAD | 29.5 | 57.6 | 2.9 | 1.9 | 11.2 |
| GeoCAD-ours | 23.8 | 62.4 | 2.3 | 1.2 | 4.3 |

## H. Visualization

Figure 6, Figure 7 and Figure 8 provide more qualitative results. When handling objects composed of multiple solids, these baseline approaches frequently struggle to infer correct arguments, resulting in disconnected components or inaccurate relative spatial configurations. In contrast, our method exhibits a stronger capability in modeling global 3D geometric relationships, enabling the generation of multiple solids that are coherently positioned according to the layout specified by the input.

Even under challenging configurations, our method generally preserves the main structural layout and successfully reconstructs the dominant geometric parts. In particular, it demonstrates clear advantages in modeling curved primitives such as arcs and circles, producing more consistent geometries than the baseline methods. This qualitative observation aligns well with the reported F1 scores for Arc and Circle in Table 1 and Table 5, further validating the effectiveness of A3PL and GCO.

## I. Limitation

Despite its overall effectiveness, our approach still exhibits several limitations. First, A3PL currently focuses on explicit numerical tokens and does not provide direct supervision for other tokens that implicitly affect geometry, such as primitive types and Boolean operations in extrusion. As revealed by the qualitative results, this often leads to errors in predicting Boolean relationships between solids, including incorrect union or subtraction behaviors. Without numerical supervision, these operations are difficult to learn solely through sequential token prediction, causing the model to generate geometrically plausible primitives that are improperly combined at the program level.

In addition, another common failure mode is the omission of minor or less salient primitives. This issue may arise from both data-related and model-related factors. On the one hand, such components are underrepresented in the training data and contribute less to global geometric metrics, resulting in weaker learning signals. On the other hand, the limited model capacity may bias generation toward major structural elements, prioritizing overall shape consistency over fine-grained completeness. As a result, for highly complex shapes or objects requiring strong combinatorial generalization, the model may only reconstruct simplified surrogate geometries.

Addressing these issues represents an important direction for future work. In particular, future research will focus on extending geometric supervision to non-numerical tokens, improving the modeling of Boolean operations, and enhancing model capacity and data coverage.

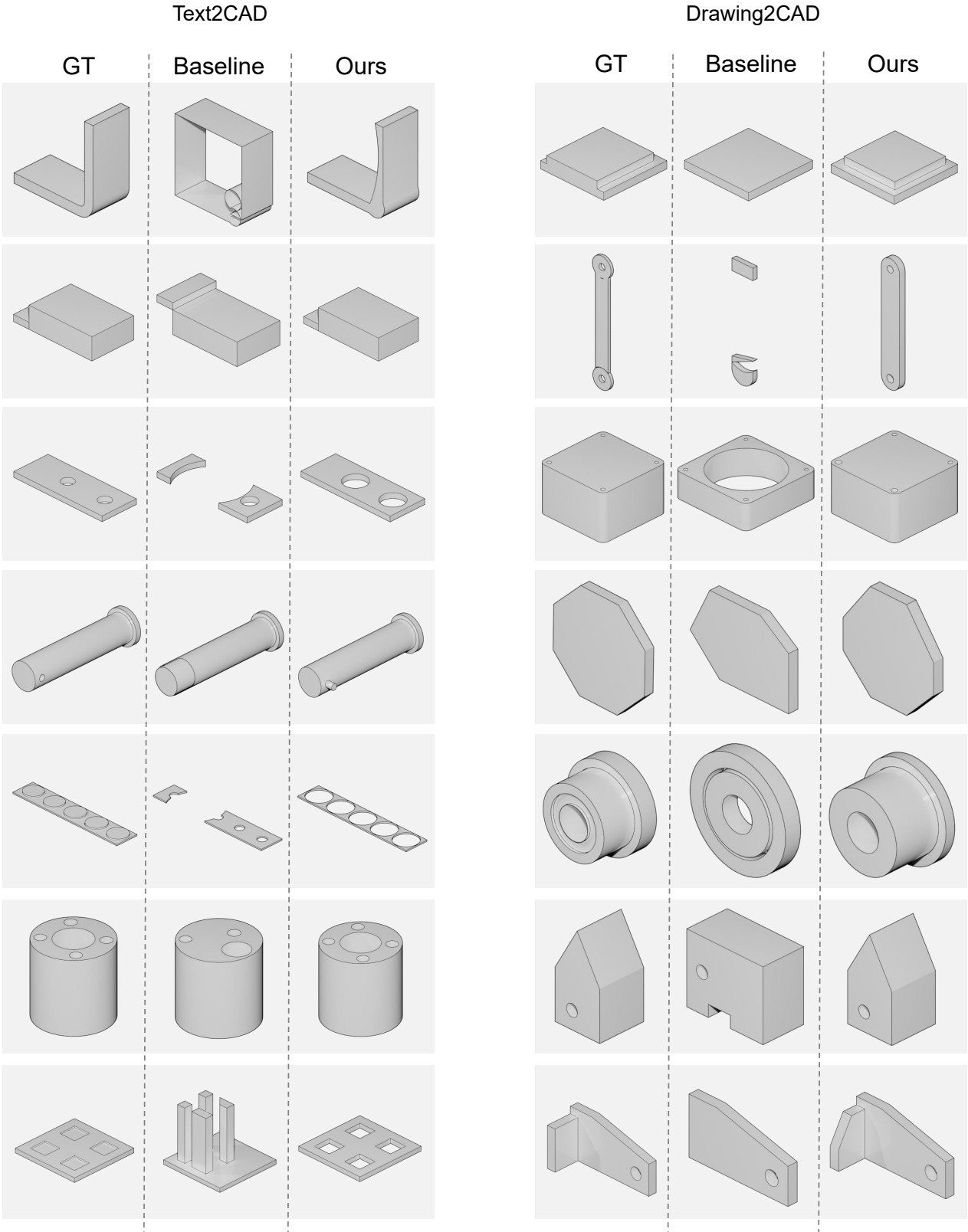

*Figure 6.* Qualitative comparisons between Text2CAD and Text2CAD-ours, as well as Drawing2CAD and Drawing2CAD-ours, corresponding to the models in Table 1. GT denotes the ground-truth CAD models.

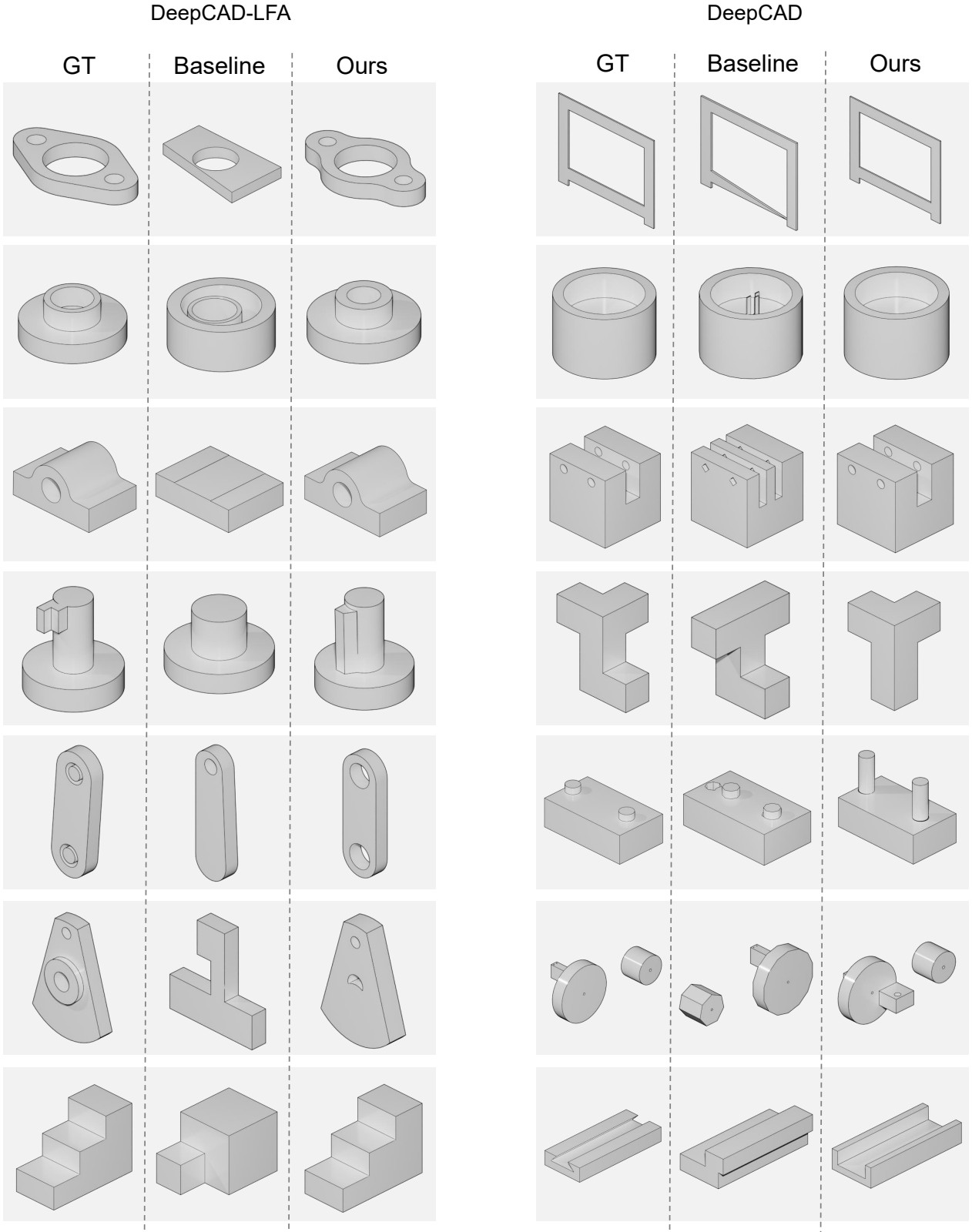

*Figure 7.* Qualitative comparisons between DeepCAD and DeepCAD-ours, as well as DeepCAD-LFA and DeepCAD-LFA-ours, corresponding to the models in Table 1. GT denotes the ground-truth CAD models.

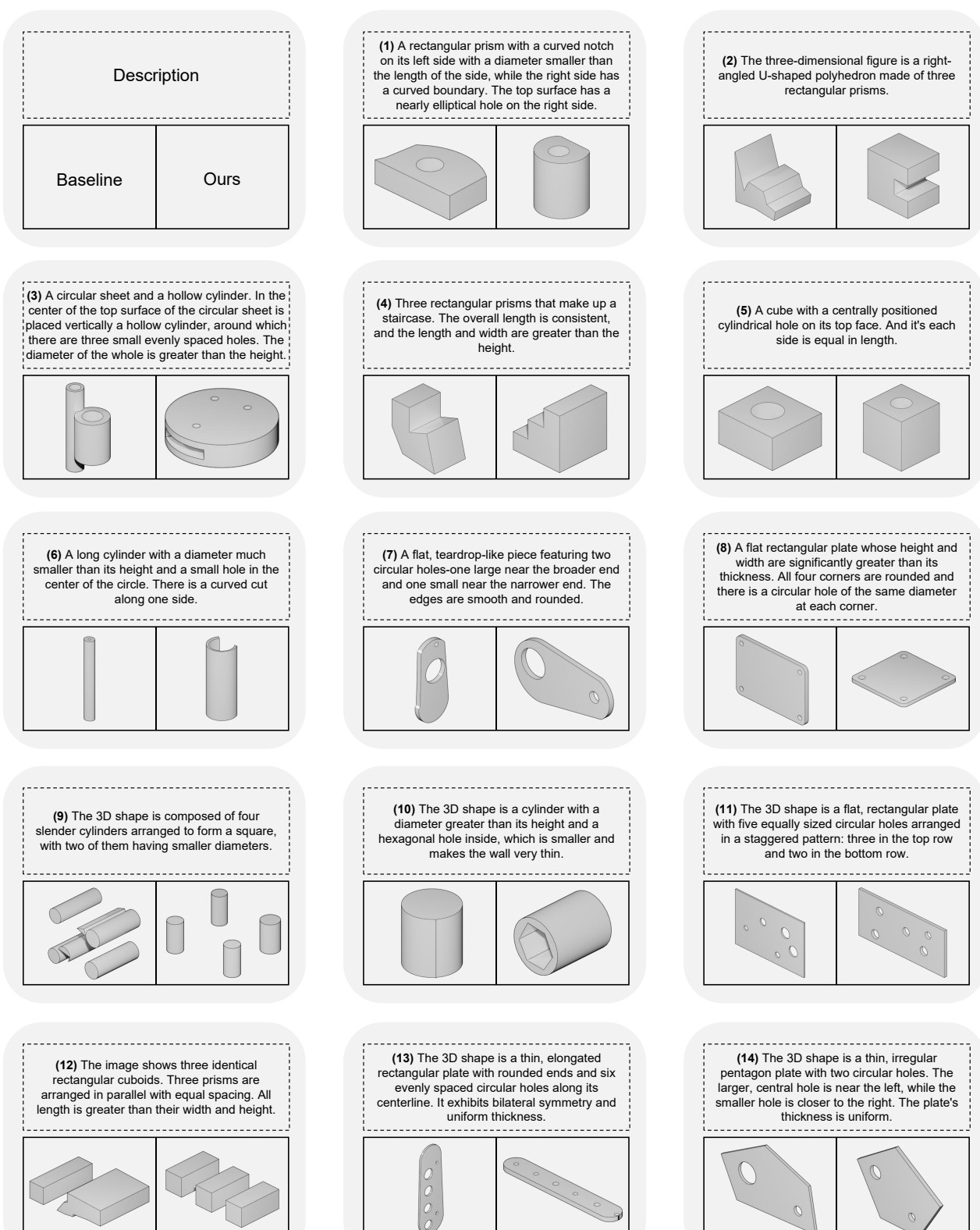

*Figure 8.* Qualitative results of CADFusion and CADFusion-ours, corresponding to the data in Table 1. The input prompt is shown at the top; outputs from the baseline CADFusion and our method are displayed in the bottom-left and bottom-right, respectively.

