# OpenReview forum: "Bridging Tokens and Geometry: Token-wise 3D Supervision for CAD Generation"
_ICML.cc/2026/Conference — ICML 2026 regular_

### Official Review · Reviewer_qszn · 2026-02-22

**Soundness:** 3
**Presentation:** 2
**Significance:** 3
**Originality:** 3
**Overall Recommendation:** 4
**Confidence:** 3

**Summary:**

This paper is about CAD sequence generation with intermediate geometric supervision. The input is modality-agnostic (text, point clouds, SVG, CAD Sequences) from 5 datasets. The method is essentially SFT with a newly designed loss function.

**Compliance With Llm Reviewing Policy:**

Affirmed.

**Final Justification:**

Thank you. Based on the current version and the rebuttal results, I have raised my rating.

**Key Questions For Authors:**

1. Initially, my biggest concern about A3PL is that it may penalize geometrically equivalent but differently transformed solutions. ESAug sounds like a possible way to mitigate this. so ESAug should not be placed in the supplementary mostly. On the other hand, some parts of GCO are not hard-core, maybe some content can be shifted.

2. Line 657: ESAug replaces **GT** extrusion params with randomly sampled valid sequences. it seems *robustness to **random** extrusion contexts improves generalization*? but can this hypothesis theoretically justified? How about not random, but more like GT with noise?

3. What is the ratio of masked tokens in GCO?

4. Line 50 right: The enumerate makes it not very pleasing to the eyes. maybe consider a different formatting style.

5. Figure 1 needs revision to be more standard. For example, what is CD? Figure 1(c) about RL is not accurate.

6. Line 133 left: *Our method … is consistent with prior works.* need citation.

7. How to precisely understand “given” (vs predicted)? it seems not GT. Please give a precise definition.

8. Line 47 left: “However, the 3D geometry is rendered after executing … geometry is difficult.” This paragraph feels disconnected from the previous logic and is quite confusing. I suggest better phrasing to improve flow.

9. nearly all headings still show “Submission and Formatting ... 2026”.

I will raise my score if the authors can convincingly address these concerns, especially #1, #2.

**Limitations:**

yes

**Strengths And Weaknesses:**

Strengths:
- A3PL and GCO as loss design are overall elegant, effective, and sounding to the CAD domain.
- The method has been tested on 5 datasets with different modalities.
- It involves different models, e.g., Transformer, llama3 8b.

Weaknesses:
- However, I think this is a niche design. The training framework here cannot be **directly** extended to more general domains, so unclear how much impact this brings to our broader AI community.

---

> ### Author Rebuttal · Authors · 2026-03-31
>
> We thank the reviewer for the patient and detailed feedback. Following your suggestions, we have revised the paper structure and improved the presentation to enhance the clarity of the manuscript.
>
> ### **Weakness: However, I think this is a niche design. The training framework here cannot be directly extended to more general domains, so unclear how much impact this brings to our broader AI community.**
>
> Thank you for raising this important question. We respectfully argue that our design is not limited to CAD alone. Instead, it targets a broader problem, namely **structured sequence generation with numerical tokens**, which arises in domains such as scene layout generation and DSL-based graphics (e.g., SVG program generation). Our method can thus be applied to the numerical tokens in these sequences.
>
> ### **Question 1: Initially, my biggest concern about A3PL is that it may penalize geometrically equivalent but differently transformed solutions. ESAug sounds like a possible way to mitigate this. so ESAug should not be placed in the supplementary mostly. On the other hand, some parts of GCO are not hard-core, maybe some content can be shifted.**
>
> Thank you for the valuable suggestion. As ESAug is important to the model's generalization ability, we make **two main revisions** to better explain it:
>
> - We move Figure 5 and the description of ESAug from Supplement **B** into **Augmentation Strategy** (Line 210 left). A theoretical demonstration related to **Question 2** is included in the supplementary material **B**.
> - We shift the details of 4 constraints (Line 247 left), dequantization (Line 121 right) and Fig.3 to new sections in supplementary material.
>
> We will include the adjustments in the revised version.
>
> ### **Question 2: Line 657: ESAug replaces GT extrusion params with randomly sampled valid sequences. It seems robustness to random extrusion contexts improves generalization? But can this hypothesis be theoretically justified? How about not random, but more like GT with noise?**
>
> We sincerely thank the reviewer for this insightful question. Due to the rebuttal word limit, we include the proof [**here**](https://github.com/anonymous-ugrange/cad-3d-token/blob/main/proof.pdf) and appreciate the reviewer’s understanding.
>
> ### **Question 3: What is the ratio of masked tokens in GCO?**
>
> Thank you for the question. The masked tokens in GCO are `<end_curve>` and `<end_extrusion>`, approximately 20% and 6% of all tokens, respectively.
>
> ### **Question 4: Line 50 right: The enumerate makes it not very pleasing to the eyes. Maybe consider a different formatting style.**
>
> Thannk you for the suggestion. We revise this with number-ordered style.
>
> ### **Question 5: Figure 1 needs revision to be more standard. For example, what is CD? Figure 1(c) about RL is not accurate.**
>
> Thank you for pointing this out. We revise Fig.1(c) to make the workflow of the RL-based method more accurate ([Figure 1](https://github.com/anonymous-ugrange/cad-3d-token/blob/main/figure%201.pdf)). The caption is updated to emphasize the differences among the three methods and provide explanations for all abbreviations.
>
> ### **Question 6: Line 133 left: "Our method ... is consistent with prior works." Need citation.**
>
> We thank the reviewer for pointing this out. We will add citations in the revised version.
>
> ### **Question 7: How to precisely understand "given" (vs predicted)? Seems not GT.**
>
> Thank you for the question. In A3PL, due to ESAug, the ground truth sequences may be replaced by randomly sampled ones. We thus used the term "given" to indicate sequences that are **deterministic inputs**. For clarity, we will **replace "given" with "ground truth" or "randomly sampled"** in the corresponding contexts.
>
> ### **Question 8: Line 47 left: "However, the 3D geometry is rendered after executing ... geometry is difficult." This paragraph feels disconnected from the previous logic and is quite confusing. I suggest better phrasing to improve flow.**
>
> Thank you for the suggestion. The revision is below in bold:
>
> CAD program sequence modeling offers a promising approach for 3D design. Its sequential and structured nature aligns well with Transformer-based architectures. **However, unlike standard sequence modeling tasks, the quality of a CAD program is not determined solely by the correctness of the token sequence; it also depends on whether the rendered 3D geometry matches the intended design. Therefore, incorporating visual information during modeling is crucial for achieving geometrically correct results.** Recent studies have incorporated visual information into CAD sequence generation.
>
> ### **Question 9: Nearly all headings still show "Submission and Formatting ... 2026".**
>
> Thank you for the suggestion. We will revise all headings to display the paper title.
>
> ### Reference
>
> [1] Bartlett, P. L., \& Mendelson, S. (2002). Rademacher and gaussian complexities: Risk bounds and structural results. Journal of machine learning research, 3(Nov), 463-482.

---

> > ### Author Rebuttal · Reviewer_qszn · 2026-04-02
> >
> > overall I think the authors provide a solid rebuttal, some issues remain though:
> >
> > > Line 626 ... visible to the autoregressive model remains unchanged ... is not fully observable during autoregressive prediction
> >
> > are there human checks in this process? How to define *visible* part precisely in CAD models especially in different views?
> >
> > > What is the ratio of masked tokens in GCO?
> >
> > more analytical data should be provided to justify the necessity of GCO.

---

> > > ### Author Response · Authors · 2026-04-05
> > >
> > > We sincerely thank you for your time and careful reconsideration of our work. We truly appreciate your valuable feedback and support throughout the review process. Wishing you all the best, and thank you again for your effort and professionalism.
> > >
> > > ### **Question 1: "Line 626 ... visible to the autoregressive model remains unchanged ... is not fully observable during autoregressive prediction" are there human checks in this process? How to define visible part precisely in CAD models especially in different views?**
> > >
> > > Thank you for the question. We would like to clarify that the term "visible" here refers to token (or sequence composed of tokens) visibility **in the autoregressive generation under the causal mask**, rather than the geometric visibility of the CAD object under different views.
> > >
> > > For A3PL, **the extrusion sequence** is required as the transformation to lift the 2D points **in the sketch sequenc**e into 3D space. Since we adopt a sketch-extrusion CAD representation, where **the extrusion sequence follows the sketch sequence** (e.g., $sketch_1$-$extrude_1$-$sketch_2$-$extrude_2$...), the computation of A3PL under the casual mask naturally involves **how the extrusion sequence is selected.** If only the ground-truth extrusion sequence is used during training, the model may overfit to specific transformation configurations. This is exactly the motivation for introducing ESAug, where the extrusion sequence is randomly sampled from valid configurations. **We hope that Fig.5 helps clearly illustrate this process.**
> > >
> > > **Therefore, no human checking is involved.** To avoid this ambiguity, we will revise the manuscript to provide a clearer and more intuitive description.
> > >
> > > ### **Question 2: What is the ratio of masked tokens in GCO? more analytical data should be provided to justify the necessity of GCO.**
> > >
> > > Thank you for the suggestion. We have further conducted an ablation study on GCO, as shown in the table below. $M_{A3PL+ESAug}$ denotes the models with A3PL and ESAug **but without GCO**. $M_{A3PL+ESAug+GCO}$ denotes the **GCO-masked** models with A3PL, ESAug, and GCO. **Each cell reports results in the order: Text2CAD / Drawing2CAD / DeepCAD-LFA.**
> > >
> > > |Method|F1 Sketch ↑|F1 Extrusion ↑|CD Mean ↓|IR ↓|
> > > |-------|-----------|--------|---------|----|
> > > |$M_{A3PL+ESAug}$|70.2 / 61.0 / 49.1|93.4 / 89.4 / 89.9|23.9 / 22.2 / 23.0|2.5 / 2.2 / 1.1|
> > > |$M_{A3PL+ESAug+GCO}$|74.8 / 61.7 / 50.8|94.9 / 89.8 / 90.1|22.7 / 21.5 / 21.6|1.9 / 1.5 / 0.8|
> > >
> > > The ablation results clearly demonstrate the necessity of GCO. After introducing GCO, **we observe consistent improvements across all four metrics under all three baselines.** In particular, the gains are most notable on F1 Sketch (e.g., from 70.2 to 74.8) and IR (e.g., from 2.5 to 1.9). Moreover, the consistent improvement in CD Mean further indicates that GCO also improves the geometric quality of the generated CAD outputs. Overall, **these findings strongly validate our motivation that masking structural tokens (GCO) effectively reduces the complexity of sequence modeling,** thereby leading to more accurate and geometrically consistent CAD generation.

---

### Official Review · Reviewer_8xpJ · 2026-03-11

**Soundness:** 4
**Presentation:** 3
**Significance:** 4
**Originality:** 3
**Overall Recommendation:** 4
**Confidence:** 4

**Summary:**

This paper introduces a new training approach for autoregressive generation of CAD models. The key idea is to impose direct token-wise 3D geometry supervision for CAD sequence prediction. The authors propose an Argument-induced 3D Point Loss (A3PL) that lifts 2D sketch points into 3D space and supervises them with the ground-truth. To ensure validity in CAD sequence generation, the authors also introduce a Grammar-constrained Operator (GCO) that uses grammar constraints to complete structural tokens automatically at inference time. Experiments show improved performance in CAD sequence generation over existing approaches.

**Compliance With Llm Reviewing Policy:**

Affirmed.

**Final Justification:**

I appreciate the authors' efforts in the rebuttal to address some of my concerns. Including the loss term ablation with better justification of the sketch loss contribution and comparison to a recent baseline could strengthen the submission. Due to these remaining issues, I maintain my original rating of weak accept.

**Key Questions For Authors:**

- Can a more complete ablation study on training losses be conducted?
- Can the missing baseline comparison be provided?

**Limitations:**

Yes

**Strengths And Weaknesses:**

Strengths:
-  Introducing direct 3D supervision to key geometric points during CAD token sequence prediction is interesting and reasonable. This process is fully differentiable. A3PL is imposed on both the sketch sequence and the extrusion sequence, guiding the network to associate CAD argument prediction with the implied 3D geometry.
- The proposed GCO operating on structural tokens ensures structural validity in CAD sequences at inference time by automatic completion according to grammars, while the structural tokens are masked out in the training loss to reduce learning complexity.
- The authors demonstrate the effectiveness and generalization of the proposed approach when compared to 5 baseline methods taking as input various conditions including text, SVG sketches, and point clouds.


Weaknesses:
- The ablation study may not be complete. The contributions of the sketch loss (Eq 3) and the extrusion loss (Eq 4) are unclear. While masking out structural tokens in the training loss reduces learning complexity, the performance difference from without masking is unclear.
- While the overall generation quality is better than the baselines, it seems the proposed method often fails to generate more fine-grained structures, for example, the last column of Fig 4.
- More recent CAD generation works, like the ones listed, are missing and worth discussion, and GeoCAD should be compared.

References
- Zhang et al. GeoCAD: Local Geometry-Controllable CAD Generation with Large Language Models. NeurIPS 2025.
- Zhang et al. FlexCAD: Unified and Versatile Controllable CAD Generation with Fine-tuned Large Language Models. ICLR 2025.

---

> ### Author Rebuttal · Authors · 2026-03-31
>
> We sincerely thank the reviewer for the insightful comments.
>
> ### **Question 1.1: The ablation study may not be complete. The contributions of the sketch loss (Eq 3) and the extrusion loss (Eq 4) are unclear.**
>
> We thank the reviewer for this suggestion. To better clarify the contributions of the sketch loss and the extrusion loss, we conduct additional experiments on the three baselines with different input modalities: Text2CAD, Drawing2CAD, and DeepCAD-LFA.
>
> In the table below, $M_{skt+ESAug}$ denotes the models with sketch loss and ESAug. $M_{ext+ESAug}$ denotes the models with extrusion loss and ESAug. $M_{A3PL+ESAug}$ denotes the models with **A3PL (skt+ext)** and ESAug. **Each cell reports results in the order: Text2CAD / Drawing2CAD / DeepCAD-LFA.**
>
> |Method|F1 Sketch ↑|F1 Extrusion ↑|CD Mean ↓|IR ↓|
> |-------|-----------|--------|---------|----|
> |$M_{skt+ESAug}$|70.0 / 61.0 / 49.1|93.1 / 89.3 /89.9|24.3 / 22.5 / 23.5|2.5 / 2.5 / 1.2|
> |$M_{ext+ESAug}$|69.9 / 60.8 / 48.9|93.3 / 89.4 /89.9|24.8 / 23.1 / 23.9|2.7 / 2.5 / 1.5|
> |$M_{A3PL+ESAug}$|70.2 / 61.0 / 49.1|93.4 / 89.4 /89.9|23.9 / 22.2 / 23.0|2.5 / 2.2 / 1.1|
>
> We observe that the sketch loss yields more improvements in CD and F1 Sketch, whereas the extrusion loss contributes more to F1 Extrusion, **consistent with their respective design objectives**.
>
> We will include this experiment in the revised version.
>
> ### **Question 1.2: While masking out structural tokens in the training loss reduces learning complexity, the performance difference from without masking is unclear.**
>
> Thank you for this suggestion. To clarify the effect of masking structural tokens during training, we provide a comparison between the two settings in the table below.
>
> $M_{A3PL+ESAug}$ denotes the **unmasked** models with A3PL and ESAug. $M_{A3PL+ESAug+GCO}$ denotes the **GCO-masked** models with A3PL, ESAug, and GCO. **Each cell reports results in the order: Text2CAD / Drawing2CAD / DeepCAD-LFA.**
>
> |Method|F1 Sketch ↑|F1 Extrusion ↑|CD Mean ↓|IR ↓|
> |-------|-----------|--------|---------|----|
> |$M_{A3PL+ESAug}$|70.2 / 61.0 / 49.1|93.4 / 89.4 / 89.9|23.9 / 22.2 / 23.0|2.5 / 2.2 / 1.1|
> |$M_{A3PL+ESAug+GCO}$|74.8 / 61.7 / 50.8|94.9 / 89.8 / 90.1|22.7 / 21.5 / 21.6|1.9 / 1.5 / 0.8|
>
> We observe that masking structural tokens consistently improves all four metrics, especially F1 Sketch and IR, which **validates the design of GCO to reduce learning complexity.**
>
> We will include this experiment in the revised version.
>
> ### **Question 2: While the overall generation quality is better than the baselines, it seems the proposed method often fails to generate more fine-grained structures, the last column of Fig 4.**
>
> We thank the reviewer for the careful observation in Fig.4.
>
> We respectfully argue that **our method can in fact improve the model's ability to generate fine-grained details.** As illustrated in Fig.4, compared with the baseline methods, our approach produces more visually sound shapes in fine-grained regions. Many other examples further support this observation, such as column 2 of Fig.4(a), column 1 of Fig.4(b), and columns 1, 2, and 4 of Fig.4(c). Furthermore, as shown in Fig.8, **our method can also model fine-grained spatial relationships among multiple objects.**
>
> We will include more examples in the revised version to further clarify this point.
>
> ### **Question 3: More recent CAD generation works, like the ones listed, are missing and worth discussion, and GeoCAD should be compared.**
>
> Thank you for the suggestion. FlexCAD [1] and GeoCAD [2] are recent representative studies in fine-grained controllable CAD generation. We will discuss the two works in **Related Works** (Line 82 Right) as follows:
>
> *Recent studies have further explored fine-grained controllable CAD generation. FlexCAD [1] introduces a unified framework for controllable generation across multiple CAD construction hierarchies. GeoCAD [2] further focuses on local geometry control, allowing users to modify specific local parts according to fine-grained geometric instructions. These works highlight the growing importance of fine-grained control and user-interactive design in CAD generation.*
>
> Regarding the comparison with GeoCAD, since the authors have not released the model weights or evaluation code for metrics, we are unable to train both the baseline and our improved models within one week using our available computational resources. Besides, we would need to implement the evaluation code ourselves. For these reasons, we can only update our experimental results in the later discussion phase. We sincerely hope that you can understand.
>
> ### Reference
>
> [1] Zhang, Z., Sun, S., Wang, W., Cai, D., & Bian, J. (2024). Flexcad: Unified and versatile controllable cad generation with fine-tuned large language models. arXiv preprint arXiv:2411.05823.
>
> [2] Zhang, Z., Liu, K., Liu, J., Wang, W., Lin, B., Xie, L., ... & Cai, D. (2025). Geocad: Local geometry-controllable cad generation. arXiv e-prints, arXiv-2506.

---

> > ### Author Rebuttal · Reviewer_8xpJ · 2026-04-02
> >
> > Thank you for the rebuttal, which addresses several of my concerns. The marginal performance difference when ablating the sketch loss is however a bit concerning. The comparison to a recent baseline is promised to be added later, which is understandable. I am keeping the weak accept rating.

---

> > > ### Author Response · Authors · 2026-04-05
> > >
> > > ### **Question 1: The marginal performance difference when ablating the sketch loss is however a bit concerning.**
> > >
> > > Thank you for the reply. Although the performance difference between $M_{ext+ESAug}$ and $M_{A3PL+ESAug}$ may seem marginal, this does not necessarily indicate limited usefulness. Instead, **we suggest that the improvements brought by the sketch loss partially overlap with those brought by the extrusion loss.** To more clearly investigate the contribution of each loss term, we compare each variant against the baseline models in the table below.
> > >
> > > **Each cell reports results in the order: Text2CAD / Drawing2CAD / DeepCAD-LFA.**
> > >
> > > |Method|F1 Sketch ↑|F1 Extrusion ↑|CD Mean ↓|IR ↓|
> > > |-------|-----------|--------|---------|----|
> > > |$M_{baseline}$|69.8 / - / 46.3|94.3 / - / 89.7|30.5 / 108.0 / 27.1|3.5 / 20.0 / 2.1|
> > > |$M_{skt+ESAug}$|70.0 / 61.0 / 49.1|93.1 / 89.3 /89.9|24.3 / 22.5 / 23.5|2.5 / 2.5 / 1.2|
> > > |$M_{ext+ESAug}$|69.9 / 60.8 / 48.9|93.3 / 89.4 /89.9|24.8 / 23.1 / 23.9|2.7 / 2.5 / 1.5|
> > > |$M_{A3PL+ESAug}$|70.2 / 61.0 / 49.1|93.4 / 89.4 /89.9|23.9 / 22.2 / 23.0|2.5 / 2.2 / 1.1|
> > >
> > > As shown in the table, both $M_{skt+ESAug}$ and $M_{ext+ESAug}$ achieve close performance to $M_{baseline}$ on F1 Sketch and F1 Extrusion. For CD Mean and IR, both losses show clear improvements over the baseline, with the sketch loss providing a slightly larger gain. **This indicates that both losses are effective.** For $M_{A3PL+ESAug}$ where the losses are used together, we still observe consistent, though relatively modest, improvements across all four metrics compared with using either loss alone. **This suggests that the gains brought by the two losses are partially overlapping, while still providing complementary benefits when jointly optimized.**
> > >
> > > ### **Question 2: Additional Experiments.**
> > >
> > > We conducted experiments on GeoCAD [1]. We trained GeoCAD and GeoCAD-ours with the code released. The training pipeline of GeoCAD consists of two stages. As the authors did not release the code for Stage I, we are only able to perform the Stage II training. The corresponding results are presented in the table below.
> > >
> > > |Method|CD Mean ↓|COV ↑|MMD ↓|JSD ↓|IR ↓|
> > > |-|-|-|-|-|-|
> > > |GeoCAD|29.5|57.6|2.9|1.9|11.2|
> > > |GeoCAD-ours|23.8|62.4|2.3|1.2|4.3|
> > >
> > > As shown in the results, our method consistently improves performance across all evaluation metrics, with especially notable gains in CD Mean, COV, and IR. **These results demonstrate the effectiveness of our method for fine-grained controllable CAD generation tasks.**
> > >
> > > ### Reference
> > >
> > > [1] Zhang, Zhanwei, et al. "GeoCAD: Local Geometry-Controllable CAD Generation with Large Language Models." The Thirty-ninth Annual Conference on Neural Information Processing Systems.

---

### Official Review · Reviewer_yFf2 · 2026-03-13

**Soundness:** 3
**Presentation:** 3
**Significance:** 3
**Originality:** 3
**Overall Recommendation:** 4
**Confidence:** 2

**Summary:**

This paper proposes an general method (Argument-induced 3D Point Loss (A3PL)) to improve auto-regressive CAD generation methods, which is evaluated on various baseline methods.

**Compliance With Llm Reviewing Policy:**

Affirmed.

**Key Questions For Authors:**

Please see weakness part.

**Limitations:**

Yes.

**Strengths And Weaknesses:**

**Strength**

1. The proposed method is a general solution to auto-regressive methods, which fills the gap of intermediate geometric feedback in CAD sequence training.
2. The method demonstrates strong generality and effectiveness, achieving consistent improvements across difference baselines.

**Weakness**

1. A3PL only provides supervision for explicit numerical tokens, where the non-numerical tokens can also influence the generated shape.
2. I do not quite understand the relation between the RL method in Fig.1 and the supervised training objective in Eq. 7. The RL training could still be 3D aware, which questions the necessity of the proposed method?

---

> ### Author Rebuttal · Authors · 2026-03-31
>
> We sincerely thank the reviewer for the thoughtful questions and valuable suggestions. Your feedback has helped improve the significance and clarity of our paper, and has also provided valuable inspiration for our future research.
>
> ### **Question 1: A3PL only provides supervision for explicit numerical tokens, where the non-numerical tokens can also influence the generated shape.**
>
> We thank the reviewer for this insightful question.
>
> Our work is motivated by a key challenge in CAD sequence generation: the actual effect of numerical tokens in the rendered 3D object often lacks effective supervision. **Therefore, the primary goal of this work is to address this issue of numerical tokens.** In this sense, our framework adopts a **task decomposition strategy**, where A3PL focuses on numerical tokens, while the standard autoregressive modeling objective (e.g., CE loss) is responsible for non-numerical tokens.
>
> Non-numerical tokens that mainly influence the generated shape are primitive type tokens: line, arc and circle. Interestingly, although A3PL does not directly supervise these three tokens, **we observe that it still brings improvements in the prediction of primitive type tokens.** As shown in Table 1, our method consistently improves the F1 scores of line, arc, and circle. A possible explanation is that primitive types are strongly coupled with their corresponding numerical tokens, and thus A3PL can in a way provide indirect supervision for them.
>
> We appreciate the reviewer for raising the issue of non-numerical token supervision. How to provide effective geometric supervision for these non-numerical tokens is an important problem in CAD sequence generation. We are currently exploring this direction and also hope this work can encourage more attention from the community toward this problem.
>
> ### **Question 2.1: I do not quite understand the relation between the RL method in Fig.1 and the supervised training objective in Eq.7.**
>
> Thank you for raising this question. There is no relation between the RL method in Fig.1(c) and the training objective in Eq.7. We have revised Fig.1 to make it more clear ([Figure 1](https://github.com/anonymous-ugrange/cad-3d-token/blob/main/figure%201.pdf)).
>
> Fig.1 illustrates different ways of incorporating visual information into CAD sequence generation. Specifically, **Fig.1(a) introduces our method.** As also described in **Introduction** (line 12 right),  Fig.1(b) introduces a way of additional visual modalities providing complementary visual information. Fig.1(c) presents how RL-based methods provide sequence-level geometric guidance.
>
> Eq.7 is the training objective of our method, which is applied to the **SFT stage**. Specifically, the final training objective jointly optimizes **A3PL loss** (consists of sketch loss in Eq.3 and extrusion loss in Eq.4) and **masked cross-entropy loss**.
>
> ### **Question 2.2: The RL training could still be 3D aware, which questions the necessity of the proposed method.**
>
> Thank you for this important question.
>
> The key limitation of RL-based methods lies in their **sequence-level and sparse** geometric feedback. Consequently, the actual geometric effects of individal numerical tokens in the rendered 3D object cannot be effectively guided. In contrast, A3PL provides **dense and token-wise geometric feedback** for each numerical token, which does not rely on the completeness of CAD programs.
>
> Moreover, RL-based methods and A3PL are **complementary rather than redundant**. In our framework, A3PL is applied during the SFT stage, which in turn provides a stronger initialization for subsequent RL optimization. As shown in Table 1 and Table 2, for the CADFusion baseline [1], which adopts an SFT+RL pipeline, it consistently leads to better performance to incorporate A3PL, suggesting **the necessity of our method**.
>
> ### Reference
>
> [1] Wang, R., Yuan, Y., Sun, S., & Bian, J. (2025). Text-to-cad generation through infusing visual feedback in large language models. arXiv preprint arXiv:2501.19054.

---

> > ### Author Rebuttal · Reviewer_yFf2 · 2026-04-05
> >
> > I thank the authors for their response. My concerns have been addressed.

---

> > > ### Author Response · Authors · 2026-04-05
> > >
> > > Thank you for your time and feedback. We appreciate your efforts in reviewing our manuscript.

---

### Decision · Program_Chairs · 2026-04-30

**Decision:**

Accept (regular)

**Comment:**

The paper received initially positive reviews with common concerns such as incomplete ablation and evaluation (qszn and 8xpJ) and some concerns about clarity of the exposition (e.g. qszn). These concerns were sufficiently addressed by the rebuttal as acknolwedged by all reviewers on the acceptance side.

Authors should please incorporate all the feedbacks, and also fix the reference issues. For example, the first reference "Adam, K. D. B. J. et al. A method for stochastic optimization. arXiv preprint arXiv:1412.6980, 1412(6), 2014." is incorrect.